# STAF: Sinusoidal Trainable Activation Functions for Implicit Neural Representation

## Abstract

Implicit Neural Representation (INR) has emerged as a promising method for characterizing continuous signals. This paper addresses the spectral bias exhibited by conventional ReLU networks, which hampers their ability to reconstruct fine details in target signals. We introduce Sinusoidal Trainable Activation Functions (STAF), designed to model and reconstruct diverse complex signals with high precision. STAF mitigates spectral bias, enabling faster learning of high-frequency details compared to ReLU networks. We demonstrate STAF's superiority over state-of-the-art networks such as KAN, WIRE, SIREN, and Fourier features, achieving higher accuracy and faster convergence with superior Peak Signal-to-Noise Ratio (PSNR). Our extensive experimental evaluation establishes STAF's effectiveness in improving the reconstruction quality and training efficiency of continuous signals, making them valuable for various applications in computer graphics and related fields.

## 1 Introduction

Implicit Neural Representations (INRs) mark a significant advancement in signal processing and computer vision, shifting from traditional discrete methods to continuous data mapping via neural networks, particularly Multilayer Perceptrons (MLPs). This shift allows for the handling of diverse data types and complex data relationships, transcending the limitations of grid-based systems and driving innovations in fields like computer graphics and computational photography (Mildenhall et al., 2020; Sitzmann et al., 2020; Tancik et al., 2020). INRs have been instrumental in novel view synthesis, 3D reconstruction, and addressing high-dimensional data challenges, such as rendering complex shapes and light interactions (Mildenhall et al., 2020; Sitzmann et al., 2020; Chen et al., 2021; Mescheder et al., 2019; Saragadam et al., 2022). Despite their versatility, traditional INR architectures, particularly those based on ReLU networks, encounter limitations due to spectral bias, which affects the reconstruction of fine details (Rahaman et al., 2019).

To address these challenges, we propose the **S**inusoidal **T**rainable **A**ctivation **F**unction **(STAF)**, a novel family of parametric, trainable activation functions that enhance the expressive power and performance of INRs in modeling complex signals. STAF generalizes periodic activation functions like SIREN (Sitzmann et al., 2020), which uses a single sinusoidal term with fixed phase and frequency, by introducing trainable parameters for greater flexibility. This development addresses challenges identified in earlier works regarding training networks with periodic activations (Lapedes & Farber, 1987; Parascandolo et al., 2016; Mehta et al., 2021) and expands the application of Fourier series in INRs (Gallant & White, 1988; Tancik et al., 2020; Shivappriya et al., 2021; Liao, 2020). Our findings indicate that STAF significantly improves neural network performance in high-fidelity applications like computer graphics and data compression.

Our work makes the following key contributions:

- **Novel Initialization Scheme:** We propose a mathematically rigorous initialization scheme that introduces a unique probability density function for initialization, providing a more robust foundation for training compared to methods relying on the central limit theorem and specific conditions, such as SIREN.

- **Expressive Power:** STAF significantly expands the set of potential frequencies compared to SIREN. By leveraging a general theorem based on the Kronecker product, we demonstrate a

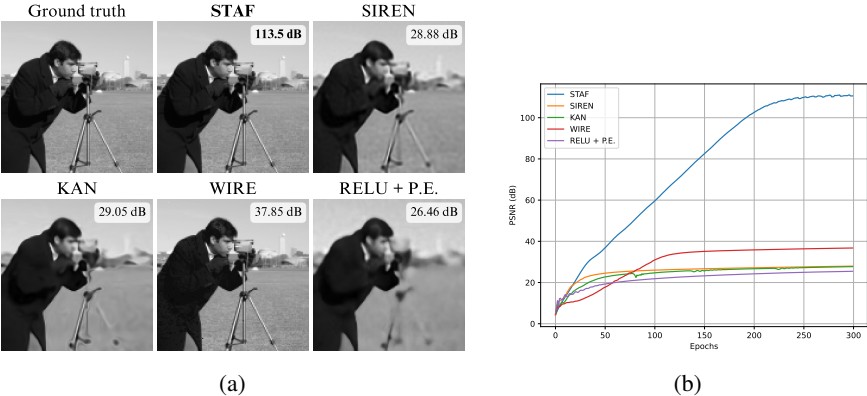

Figure 1: (a) Ground truth image followed by reconstructions using STAF, WIRE, KAN, SIREN, and ReLU + Positional Encoding. (b) PSNR values achieved over training iterations, demonstrating STAF's superior performance.

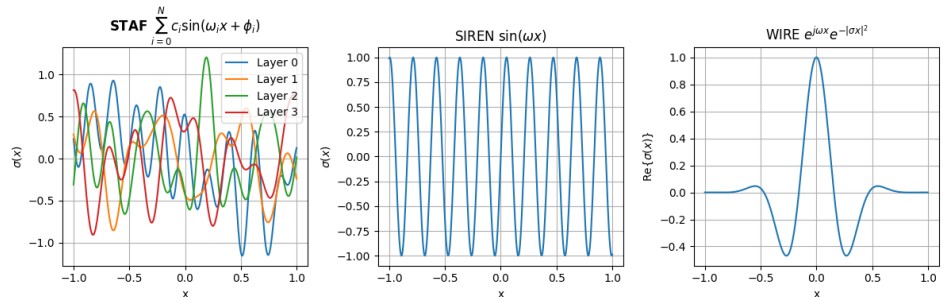

Figure 2: Activation functions used in INRs plotted over the range [-1, 1]. STAF utilizes a parameterized Fourier series activation, offering flexible frequency-domain adaptation. SIREN employs a sinusoidal function, providing a periodic activation landscape. WIRE employs a complex Gabor wavelet activation, balancing spatial and frequency localization.

substantial increase in the expressive capacity of our network. Theorems 3 and 4, which we provide, extend beyond STAF, offering novel insights into any trainable activation function. We exploit some combinatorial and algebraic tools for this purpose.

- **NTK Eigenvalues and Eigenfunctions:** We analyze the Neural Tangent Kernel (NTK) of our network, showing that its eigenvalues and eigenfunctions provide improved criteria for the learning process and convergence, enhancing understanding and performance during training.

- **Performance Improvements:** Our proposed activation function leads to significant gains in performance, notably improving Peak Signal-to-Noise Ratio (PSNR) in various tasks such as image, shape, and audio representation, as illustrated in Figures 1, 3, 4, 6, and 7. These improvements are achieved through faster convergence and greater accuracy, positioning STAF as a superior alternative to state-of-the-art models such as WIRE (Saragadam et al., 2023), SIREN (Sitzmann et al., 2020), KAN (Liu et al., 2024), Gaussian (Ramasinghe & Lucey, 2022), MFN (Fathony et al., 2020), and FFN (Tancik et al., 2020).

## 2 RELATED WORKS

INRs have advanced in representing various signals, including images and 3D scenes, with applications in SDFs, audio signals, and data compression. Sitzmann et al.'s sine-based activations in INRs (Sitzmann et al., 2020) improved fidelity but faced slow training. Dual-MLP architectures (Mehta et al., 2021), input division into grids (Aftab et al., 2022; Kadarvish et al., 2021), and adap-

tive resource allocation (Martel et al., 2021) further enhanced INR capabilities. Mildenhall et al.'s volume rendering for 3D scene representation in NeRF (Mildenhall et al., 2020) inspired subsequent enhancements (Martin-Brualla et al., 2021; Barron et al., 2023; Kazerouni et al., 2024; Xu et al., 2023; Srinivasan et al., 2021; Zhang et al., 2020; Neff et al., 2021; Reiser et al., 2021) for improved fidelity and expedited rendering.

The development of neural networks has been significantly influenced by the development of activation functions. Early non-periodic functions like sigmoid faced vanishing gradient issues in deep networks, addressed by unbounded functions like ReLU (Nair & Hinton, 2010) and its variants ((Maas et al., 2013; Elfwing et al., 2018; Hendrycks & Gimpel, 2016)). Adaptive functions like SinLU (Paul et al., 2022), TanhSoft (Biswas et al., 2021), and Swish ((Ramachandran et al., 2017)) introduced trainable parameters for adapting to data non-linearity. However, spectral bias in ReLU-based networks, as highlighted by Rahaman et al. (Rahaman et al., 2019), led to a preference for low-frequency signals. Periodic activation functions emerged as promising in INRs for learning high-frequency details. Early challenges in training networks with periodic activations (Lapedes & Farber, 1987; Parascandolo et al., 2016) were overcome by successful applications in complex data representation (Sitzmann et al., 2020; Mehta et al., 2021). Fourier Neural Networks (FFN), introduced by Galant and White (Gallant & White, 1988), and Tancik et al.'s FFN with Fourier feature mapping (Tancik et al., 2020) further explored Fourier series in neural networks. This research informed the development of a parametric periodic activation function for MLP-based INR structures, targeting enhanced convergence and detail capture.

Recently, the Kolmogorov-Arnold Network (KAN) (Liu et al., 2024; SS et al., 2024) has emerged as a promising architecture in the realm of INRs. KAN leverages Kolmogorov-Arnold representation frameworks to improve the modeling and reconstruction of complex signals, demonstrating notable performance in various INR tasks. However, as we will demonstrate in our experimental results, STAF outperforms KAN in terms of accuracy, convergence speed, and PSNR. This highlights the superior capability of STAF in capturing high-frequency details and achieving higher fidelity in signal representation.

# 3 STAF: SINUSOIDAL TRAINABLE ACTIVATION FUNCTION

## 3.1 INR PROBLEM FORMULATION

INRs utilize MLPs to revolutionize traditional data representation and processing techniques. At the core of INR is the function $f_{\boldsymbol{\theta}} : \mathbb{R}^{F_0} \to \mathbb{R}^{F_L}$, where $F_0$ and $F_L$ represent the dimensions of the input and output spaces, respectively, and $\boldsymbol{\theta}$ denotes the parameters of the MLP. The objective is to approximate a target function $g(\boldsymbol{x})$ such that $g(\boldsymbol{x}) \approx f_{\boldsymbol{\theta}}(\boldsymbol{x})$. For example, in image processing, $g(\boldsymbol{x})$ could be a function mapping pixel coordinates to their respective values.

As mentioned in (Yüce et al., 2022), the majority of INR architectures can be decomposed into a mapping function $\gamma : \mathbb{R}^D \to \mathbb{R}^T$ followed by an MLP, with weights $\boldsymbol{W}^{(l)} \in \mathbb{R}^{F_l \times F_{l-1}}$ and activation function $\rho^{(l)} : \mathbb{R} \to \mathbb{R}$, applied element-wise at each layer $l = 1, \ldots, L-1$. In other words, if we represent $\boldsymbol{z}^{(l)}$ as the post-activation output of each layer, most INR architectures compute

$$
\begin{aligned}
\boldsymbol{z}^{(0)} &= \gamma(\boldsymbol{r}), \\
\boldsymbol{z}^{(l)} &= \rho^{(l)}(\boldsymbol{W}^{(l)}\boldsymbol{z}^{(l-1)} + \boldsymbol{B}^{(l)}), \quad l = 1, ..., L-1, \\
f_{\boldsymbol{\theta}}(\boldsymbol{r}) &= \boldsymbol{W}^{(L)}\boldsymbol{z}^{(L-1)} + \boldsymbol{B}^{(L)}.
\end{aligned}
\tag{1}
$$

Additionally, corresponding to the $i$'th neuron of the $l$'th layer, we employ the symbols $a_i^{(l)}$ and $z_i^{(l)}$ for the pre-activation and post-activation functions respectively. The choice of the activation function $\rho$ is pivotal in INR, as it influences the network's ability to represent signals. Traditional functions, such as ReLU, may not effectively capture high-frequency components. The novel parametric periodic activation function, i.e., STAF, enhances the network's capability to accurately model and reconstruct complex, high-frequency signals.

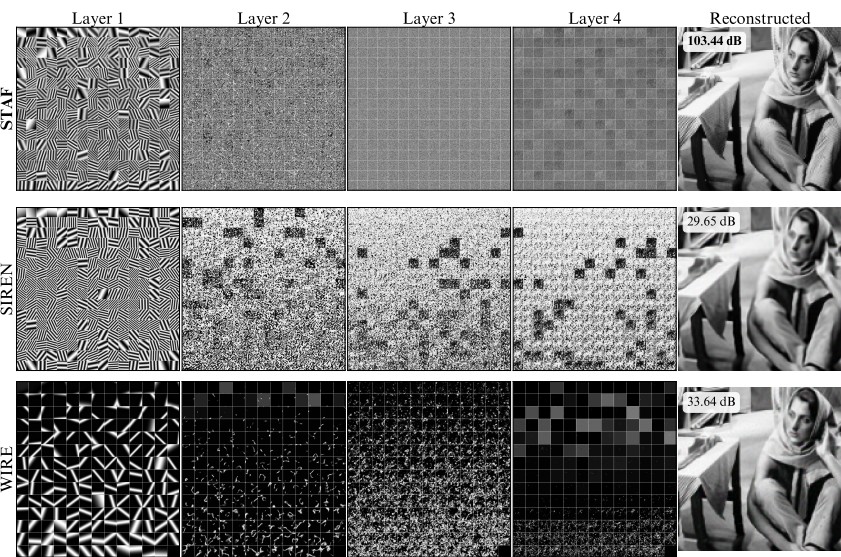

Figure 3: Activation maps for STAF, SIREN and WIRE learned during the image reconstruction task.

## 3.2 STAF ACTIVATION FUNCTION

The activation function STAF is conceptually distinct from conventional activation functions. It is parameterized, similar to a Fourier series:

$$\rho^*(x) = \sum_{i=1}^{\tau} C_i \sin(\Omega_i x + \Phi_i) \tag{2}$$

where $C_i$, $\Omega_i$, and $\Phi_i$ are the amplitude, frequency, and phase parameters of the series, respectively. These parameters are dynamically learned during the training process, allowing the network to adaptively optimize its activation function based on the specific characteristics of the signal being processed. The rationale behind using a Fourier series is its proven efficiency in capturing the energy of a signal with a minimal number of coefficients, thus allowing for a more compact and expressive representation of complex patterns.

## 3.3 STAF TRAINING PROCESS

During training, STAF optimizes not only the traditional MLP parameters (weights and biases), but also the coefficients of the activation function. This dual optimization approach ensures that the network learns both an optimal set of transformations (through weights and biases) and an ideal way of activating neurons (through the parametric activation function) for each specific task. The training employs a loss function designed to minimize the difference between the target function $g(\boldsymbol{x})$ and the network's approximation $f_{\boldsymbol{\theta}}(\boldsymbol{x})$, while also encouraging efficient representation inspired by Fourier series.

## 3.4 IMPLEMENTATION STRATEGIES

The implementation of STAF's parametric activation functions can be approached in three ways:

❶ **Individual Neuron Activation:** This method assigns a unique activation function to each neuron. It offers high expressiveness, but leads to a significant increase in the number of trainable parameters, making it impractical for large networks due to potential overfitting and computational inefficiencies.

❷ **Uniform Network-wide Activation:** Here, a single shared activation function is used across the entire network. This approach simplifies the model by reducing the number of additional parameters but limits the network's expressiveness and adaptability. It may struggle to capture diverse patterns and details in complex signals.

❸ **Layer-wise Shared Activation:** This balanced strategy employs a distinct shared activation function for each layer which is also used for all experiments in this paper. For example, in a 3-layer

MLP with $\tau = 25$ terms, only 225 additional parameters are required. This approach optimally balances expressiveness and efficiency, allowing each layer to develop specialized activation dynamics for the features it processes. It aligns with the hierarchical nature of MLPs, where different layers capture different signal abstractions, providing an efficient learning mechanism tailored to each layer's role.

### 3.5 INITIALIZATION

In this section, we examine how to initialize a network that uses STAF as its activation function. Since STAF is similar to SIREN (Sitzmann et al., 2020), which uses $\sin$ as the activation function, we compare our initialization scheme with the one used for SIREN.

Let's examine some important points regarding the initialization of SIREN, as discussed in (Sitzmann et al., 2020). In this approach, the input $X$ of a single neuron follows a uniform distribution $U(-1, 1)$, and the activation function employed is $\rho(u) = \sin(u)$. Consequently, the output of the neuron is given by $Y = \sin(aX + b)$, where $a, b \in \mathbb{R}$. The authors of (Sitzmann et al., 2020) claim that regardless of the choice of $b$, if $a > \frac{\pi}{2}$, the output $Y$ follows an arcsine distribution, denoted as $Arcsine(-1, 1)$. However, it becomes apparent that this claim is not correct upon further examination. If the claim were true, $\mathbb{E}[Y]$ would be independent of $b$. Let's calculate it in a more general case, where instead of the interval $[-1, 1]$, we consider an arbitrary interval $[c, d]$ for the input $X$.

$$\mathbb{E}[Y] = \int_c^d \sin(ax + b) f_X(x)\, dx = \frac{1}{d - c} \int_c^d (\sin(ax)\cos b + \sin b \cos(ax))\, dx$$

$$= \frac{1}{a(d - c)} \left[ (\cos(ac) - \cos(ad))\cos b + (\sin(ad) - \sin(ac))\sin b \right]. \tag{3}$$

Assuming $c = -1$ and $d = 1$, the result will be $\frac{2\sin a \sin b}{a(d-c)}$, which obviously depends on $a$ and $b$. However, if we want to eliminate $b$ from $\mathbb{E}[Y]$, we can set $ad = ac + 2n\pi$, or equivalently

$$d = c + \frac{2n\pi}{a}, \tag{4}$$

for an $n \in \mathbb{N}$. Next, let us consider the next moments of $Y$, because if the moment generating function (MGF) of $Y$ exists, the moments can uniquely determine the distribution of $Y$.

$$\mathbb{E}[Y^k] = \int_c^d \frac{\sin^k(ax + b)}{d - c} dx \tag{5}$$

Using equation 4, it is equal to

$$\frac{1}{2n\pi} \int_c^{c + \frac{2n\pi}{a}} \sin^k(ax + b) dx \tag{6}$$

By assuming $u = ax + b$, we have

$$\mathbb{E}[Y^k] = \frac{1}{2an\pi} \int_{ac+b}^{ac+b+2n\pi} \sin^k(u) du. \tag{7}$$

Since for each pair of natural numbers $(k, n)$, $2n\pi$ is a period of $\sin^k(u)$, we can write

$$\mathbb{E}[Y^k] = \frac{1}{2an\pi} \int_0^{2\pi} \sin^k(u) du = \begin{cases} 0, & \text{if } k \text{ is odd} \\ \frac{\binom{k}{k/2}}{2^k an}, & \text{if } k \text{ is even} \end{cases} \tag{8}$$

As you can see, even in this case, the moments of $Y$ (and thus the distribution of $Y$) depend on $a$ (the weight multiplied by the input) and $n$ (a parameter defining the range of input).

In the subsequent parts of (Sitzmann et al., 2020), the authors utilized the assumption that the outputs of the first layer follow an arcsine distribution and fed those outputs into the second layer. By relying on the central limit theorem (CLT), they demonstrated that the output of the second layer, for each neuron, conforms to a normal distribution. Additionally, in Lemma 1.6, they established that if $X \sim \mathcal{N}(0, 1)$ and $Y = \sin(\frac{\pi}{2}X)$, then $Y \sim Arcsine(-1, 1)$. However, it should be noted that to prove this result, they relied on several approximations. Through induction, they asserted that the

Figure 4: Comparative visualization of image representation with **STAF** and other activation functions. In the second row, we demonstrate the representation errors of different models. The brighter areas indicate higher representation errors.

inputs of subsequent layers follow an arcsine distribution, while the outputs of these layers exhibit a normal distribution.

In contrast to the approach taken by (Sitzmann et al., 2020), the method presented in this study does not depend on the specific distributions of the input vector $r$ and weight matrices $W^{(l)}$. As a result, there is no need to map the inputs to the interval $[-1, 1]$. Additionally, this method does not rely on making any approximations or the central limit theorem, which assumes large numbers. Overall, it offers a more rigorous mathematical framework. To pursue this goal, notice the following theorem.

**Theorem 1.** *Consider a neural network as defined in equation 1 with a sinusoidal trainable activation function (STAF) defined in equation 2. Suppose for each $i$, $\Phi_i \sim U(-\pi, \pi)$. Furthermore, let $C_i$ be i.i.d. random variables with the following probability density function:*

$$f_{C_i}(c_i) = \frac{\tau |c_i|}{2} e^{\frac{-\tau c_i^2}{2}}, \tag{9}$$

*and assume that $C_i$'s are independent of $\Omega_i$, $w$, $x$, and $\Phi_i$. Then, every post-activation will follow a $\mathcal{N}(0, 1)$ distribution (Please refer to the proof in Appendix C.1.).*

This initial setting, where every post-activation follows a standard normal distribution, is beneficial because it prevents the post-activation values from vanishing or exploding. This ensures that the signals passed from layer to layer remain within a manageable range, particularly in the first epoch. The first epoch is crucial as it establishes the foundation for subsequent learning. If the learning process is well-posed and there is sufficient data, the training process is likely to converge to a stable and accurate solution. Therefore, while it is important to monitor for potential issues in later epochs, the concern about vanishing or exploding values is significantly greater during the initial stages. Proper initialization helps mitigate these risks early on, facilitating smoother and more effective training overall.

## 4 EXPERIMENTAL RESULTS

We evaluated various neural network models for image reconstruction using a standard architecture across all experiments. Specifically, we employed a three-layer MLP with nonlinear activation functions in the hidden layers and a linear activation in the output layer, mirroring the structure and hyperparameters from (Sitzmann et al., 2020). Each hidden layer consisted of 256 features. The models tested included WIRE, Gauss, SIREN with positional encoding, STAF, and MFN (Saragadam et al., 2023; Sitzmann et al., 2020; Tancik et al., 2020; Fathony et al., 2020; Ramasinghe & Lucey, 2022). We also provided comparison with the recently published KAN networks (Liu et al., 2024) and in particular used Chebyshev-Polynomial KAN which offers more efficient implementation of KAN networks (SS et al., 2024). All experiments were conducted on a desktop PC equipped with 32 GB of RAM and an NVIDIA RTX-3090 GPU. Our implementation was inspired by the codebases of SIREN (https://github.com/vsitzmann/siren) and WIRE (https://github.com/vishwa91/wire/tree/main). Due to GPU resource constraints, images were resized to $128 \times 128$ pixels. This reduction ensured manageable computational demands while preserving enough detail for meaningful reconstruction analysis.

**Learning Rate Configurations** The learning rates for each model were selected based on the optimal configurations reported in their respective original papers. For WIRE, the best performance was achieved with a learning rate of $5 \times 10^{-3}$, Gauss, SIREN, ReLU with Positional Encoding, and MFN demonstrated optimal results at learning rates of $1 \times 10^{-3}$, $1 \times 10^{-4}$, $5 \times 10^{-4}$, and $1 \times 10^{-2}$, respectively. For STAF, we adopted the learning rate configuration used for the SIREN model, which was $1 \times 10^{-4}$. All models utilized the Adam optimizer during the training process to ensure consistency in optimization and comparison.

**Model Initialization** STAF was initialized using the methodology described in Section 3.5 of our paper, which is tailored to enhance model convergence and performance. For other models, we followed the initialization strategies recommended in their respective original papers, ensuring optimization according to best practices identified in prior research.

**Results and Analysis** Figures 1a and 1b illustrate the performance comparison. Figure 1a shows the ground truth image and reconstructions using STAF, WIRE, KAN, SIREN, and ReLU with Positional Encoding. Figure 1b presents the PSNR values achieved over training iterations, demonstrating STAF's superior performance.

Figure 8b plots the activation functions used in INRs over the range [-1, 1]. STAF utilizes a parameterized Fourier series activation, offering flexible frequency-domain adaptation. SIREN employs a sinusoidal function for a periodic activation landscape, while WIRE uses a complex Gabor wavelet activation, balancing spatial and frequency localization.

Figure 3 shows activation maps learned during the image reconstruction task. STAF produces more detailed and higher-quality reconstructions compared to SIREN and WIRE, highlighting its ability to capture complex features more effectively.

Figure 4 compares the PSNR achieved by different models during the image reconstruction task. The ground truth image had a PSNR of 104.57 dB. STAF achieved 37.65 dB, outperforming SIREN (51.41 dB), WIRE (70.13 dB), MFN (53.93 dB), Gaussian (41.69 dB), and FFN (22.51 dB).

The reconstructed images and the progression of PSNR values during training provide insight into each model's capabilities. STAF emerged as the leading model, achieving the highest PSNR, indicative of its superior ability to reconstruct images with greater clarity and detail. We have also conducted experiments on different signals, including shape and audio (see Appendix B), and provided a detailed NTK analysis (see Appendix A) of our model in the Appendix.

**Discussion** While the main focus of this paper is the introduction and theoretical justification of STAF, our experimental results substantiate its practical efficacy. STAF is less sensitive to weight initialization compared to SIREN, though hyperparameter tuning is still required for different tasks. This requirement could be viewed as a limitation; however, our primary emphasis remains on the theoretical analysis. Overall, STAF demonstrates a significant improvement in image reconstruction tasks, both in terms of convergence speed and reconstruction quality, making it a significant contribution to the toolkit for implicit neural representation in computer graphics and related fields.

## 5 EXPRESSIVE POWER

In this part, we examine the expressive power of our architecture, drawing upon the notable Theorem 1 from (Yüce et al., 2022). This theorem is as follows:

**Theorem 2.** *(Theorem 1 of (Yüce et al., 2022)) Let $f_{\boldsymbol{\theta}} : \mathbb{R}^D \to \mathbb{R}$ be an INR of the form of Equation equation 1 with $\rho^{(l)}(x) = \sum_{j=0}^{J} \alpha_j x^j$ for $l > 1$. Furthermore, let $\boldsymbol{\Psi} = [\boldsymbol{\Psi}_1, ..., \boldsymbol{\Psi}_T]^{tr} \in \mathbb{R}^{T \times D}$ and $\boldsymbol{\zeta} \in \mathbb{R}^T$ denote the matrix of frequencies and vector of phases, respectively, used to map the input coordinate $r \in \mathbb{R}^D$ to $\gamma(r) = \sin(\boldsymbol{\Psi} \boldsymbol{r} + \boldsymbol{\zeta})$. This architecture can only represent functions of the form*

$$f_{\boldsymbol{\theta}}(r) = \sum_{\boldsymbol{w'} \in \mathcal{H}(\Psi)} c_{w'} \sin(\langle \boldsymbol{w'}, \boldsymbol{r} \rangle + \zeta_{\boldsymbol{w'}}),$$

*where*

$$\mathcal{H}(\boldsymbol{\Psi}) \subseteq \tilde{\mathcal{H}}(\boldsymbol{\Psi}) = \left\{ \left. \sum_{t=1}^{T} s_t \boldsymbol{\Psi}_t \right| s_t \in \mathbb{Z} \wedge \sum_{t=1}^{T} |s_t| \leq J^{L-1} \right\}.$$

Please note the following remarks regarding this theorem:

**Remark 1.** *We refer to $\tilde{\mathcal{H}}$ as the set of potential frequencies.*

**Remark 2.** *The expression $\sum_{t=1}^{T} s_t \boldsymbol{\Psi}_t$ is equal to $\boldsymbol{\Psi}^{tr}[s_1, ..., s_T]^{tr}$. This representation is more convenient for our subsequent discussion, as we will be exploring the kernel of $\boldsymbol{\Psi}$ in the sequel.*

**Remark 3.** *In the context of SIREN, where $\rho^{(l)} = \sin$, the post-activation function of the first layer, $z^{(0)} = \sin(\omega_0(\boldsymbol{W}^{(0)}\boldsymbol{r} + \boldsymbol{b}^{(0)}))$, can be interpreted as $\gamma(\boldsymbol{r}) = \sin(\boldsymbol{\Psi}\boldsymbol{r} + \boldsymbol{\zeta})$.*

We will now investigate the significant enhancement in expressive power offered by the proposed activation function. To facilitate comparison with SIREN, we express our network using $\sin$ as the activation function.

Let us consider a neural network with a parametric activation function defined in equation 2. To represent our network using SIREN, we demonstrate that every post-activation function of our network from the second layer onwards ($z^{l+1}$) can be expressed using linear transformations and sine functions. Notably, the final post-activation function ($z^{(L-1)}$) can be constructed using SIREN, albeit requiring more neurons than STAF. In other words, our network can be described using a SIREN and some Kronecker products denoted by $\otimes$. This analysis resembles that provided in (Jagtap et al., 2022), with a slight difference in the settings of the paper. In (Jagtap et al., 2022), it was shown that an adaptive activation function of the form

$$\rho^*(x) = \sum_{i=1}^{\tau} C_i \rho_i(\Omega_i x) \tag{10}$$

can be represented using a feed-forward neural network, where each layer has neurons with activation functions $\rho_i$. To align STAF with this theorem, we must have $\rho_i = \sin(\Omega_i x + \Phi_i)$. However, here we aim to represent STAF using an architecture that only employs sine activation functions (SIREN). For this purpose, we introduce the following theorem, which holds true for every parametric activation function:

**Theorem 3.** *Let $L \geq 2$ and $1 \leq l \leq L$. Consider a neural network as defined in equation 1 with $L$ layers. In addition, let $\boldsymbol{\Omega} = [\Omega_1, ..., \Omega_\tau]^{tr}$, $\boldsymbol{\Phi} = [\Phi_1, ..., \Phi_\tau]^{tr}$, and $\boldsymbol{C} = [C_1, ..., C_\tau]^{tr}$. If the trainable activation function is $\rho^*(x) = \sum_{m=1}^{\tau} \boldsymbol{C}_m \rho(\boldsymbol{\Omega}_m x + \boldsymbol{\Phi}_m)$, then an equivalent neural network with activation function $\rho(x)$ and $L + 1$ layers can be constructed as follows (parameters of the equivalent network are denoted with an overline):*

$$\overline{z^{(0)}} = \gamma(\boldsymbol{r}),$$
$$\overline{z^{(l)}} = \rho\left(\overline{\boldsymbol{W}^{(l)}}\,\overline{z^{(l-1)}} + \overline{\boldsymbol{B}^{(l)}}\right), \quad l = 1, ..., L, \tag{11}$$
$$\overline{f_{\overline{\theta}}}(\boldsymbol{r}) = \overline{\boldsymbol{W}^{(L+1)}}\,\overline{z^{(L)}};$$

*where*

$$\overline{\boldsymbol{W}^{(l)}} = \begin{cases} \boldsymbol{\Omega} \otimes \boldsymbol{W}^{(l)}, & \text{if } l = 1 \\ (\boldsymbol{\Omega} \otimes \boldsymbol{C}^{tr}) \otimes \boldsymbol{W}^{(l)}, & \text{if } l \text{ is even} \\ (\boldsymbol{\Omega} \otimes \boldsymbol{W}^{(l)})(\boldsymbol{C}^{tr} \otimes \boldsymbol{I}_{F_{l-1}}), & \text{if } l \text{ is odd, } l > 1, \text{ and } l \neq L+1 \\ \boldsymbol{C}^{tr} \otimes \boldsymbol{I}_{F_{l-1}}, & \text{if } l \text{ is odd, } l > 1, \text{ and } l = L+1 \end{cases}, \quad \text{and} \quad \overline{\boldsymbol{B}^{(l)}} = \boldsymbol{\Phi} \otimes \boldsymbol{J}_{F_l};$$

$$\tag{12}$$

*in which $\boldsymbol{J}_{F_l}$ is an all-ones $F_l \times 1$ vector. Furthermore, if $L$ is even, then $\overline{f_{\overline{\theta}}}(\boldsymbol{r}) = f_\theta(\boldsymbol{r})$ (we call these networks 'Kronecker equivalent' in this sense).*

The proof of this theorem is provided in the Appendix C.2. As we observed, although a network with the activation function $\rho^*$ can be represented using the activation function $\rho$, it features a unique architecture. These networks are not merely typical MLPs with the activation function $\rho$, as the weights in the Kronecker equivalent network exhibit dependencies due to the Kronecker product.

It is desirable that Theorem equation 3 does not depend on the parity of $L$. To achieve this, consider the following remark:

**Remark 4.** *We can introduce a dummy layer with the activation function $\rho^*$. Specifically, we define $\boldsymbol{z}^{(L)} = \rho^*\left(f_\theta(\boldsymbol{r})\right)$, and $\tilde{f}_\theta(\boldsymbol{r}) = \boldsymbol{W}^{(L+1)}\boldsymbol{z}^{(L)} + \boldsymbol{B}^{(L+1)}$, where $\boldsymbol{W}^{(L+1)} = \boldsymbol{O}$. To ensure that $\tilde{f}_\theta(\boldsymbol{r}) = f_\theta(\boldsymbol{r})$, we set $\boldsymbol{B}^{(L+1)} = f_\theta(\boldsymbol{r})$. This approach allows us to construct an equivalent neural network with one more layer.*

As a result of Remark equation 4, the equivalent network of a network with trainable activation function, has either one more layer, or the same number of layers.

As an immediate result of Theorem equation 3, if we denote the embedding of the first layer of the SIREN equivalent of our network by $\overline{\boldsymbol{\Psi}}$, then

$$\overline{\boldsymbol{\Psi}} = \overline{\boldsymbol{W}^{(1)}} = \boldsymbol{\Omega} \otimes \boldsymbol{W}^{(1)} \in \mathbb{R}^{\tau F_1 \times F_0} \tag{13}$$

which is $\tau$ times bigger than the embedding of the first layer of a SIREN with $\boldsymbol{W}^{(1)} \in \mathbb{R}^{F_1 \times F_0}$. To understand the impact of this increase on expressive power, it suffices to substitute $T$ with $\tau T$ in Theorem equation 2. The next theorem will reveal how this change will affect the cardinality of the set of potential frequencies.

**Theorem 4.** *(Page 4 of (Kiselman, 2012)) Let $V(T,K) = \left\{(s_1, s_2, \ldots, s_T) \in \mathbb{Z} \;\middle|\; \sum_{t=1}^{T} |s_t| \leq K\right\}$.[1] Then we have*

$$|V(T,K)| = \sum_{i=0}^{min(K,T)} \binom{i}{K}\binom{i}{T} 2^i \tag{14}$$

*This number is called Delannoy number. Moreover, for fixed $K$,*

$$|V(T,K)| \sim A_K(2T)^K, \quad T \to +\infty \tag{15}$$

As an immediate result of this theorem, for large values of $T$,

$$\frac{|V(\tau T, K)|}{|V(T, K)|} \sim \tau^K \tag{16}$$

Now, it is time to analyze the cardinality of the set of potential frequencies:

$$\tilde{\mathcal{H}}(\boldsymbol{\Psi}) = \left\{ \sum_{t=1}^{T} s_t \boldsymbol{\Psi}_t \;\middle|\; (s_1, s_2, \ldots, s_T) \in V(T, J^{L-1}) \right\} \tag{17}$$

or equivalently,

$$\tilde{\mathcal{H}}(\boldsymbol{\Psi}) = \left\{ \boldsymbol{\Psi}^{tr}[s_1, ..., s_T]^{tr} \;\middle|\; s_t \in \mathbb{Z} \wedge \sum_{t=1}^{T} |s_t| \leq J^{L-1} \right\} \tag{18}$$

The cardinality of the set $\tilde{\mathcal{H}}(\boldsymbol{\Psi})$ is bounded above by $V(T, J^{L-1})$. If $\boldsymbol{\Psi}^{tr}$, is injective on the integer lattice $\mathbb{Z}^T$, then $|\tilde{\mathcal{H}}(\boldsymbol{\Psi})| = |V(T, J^{L-1})|$. However, in general, analyzing how a linear transformation affects the size of a convex body can be approached using the geometry of numbers (Matousek, 2013) or additive geometry (Tao & Vu, 2006). To simplify the analysis and preserve the size of $\tilde{\mathcal{H}}(\boldsymbol{\Psi})$ as large as possible, we can slightly perturb the matrix $\boldsymbol{\Psi}^{tr}$ such that its kernel contains no points with rational coordinates, except the origin. This is a much stronger condition than having no integer lattice points in the kernel. To address this, we introduce a lemma. It's worth noting that we can assume the matrices are stored with rational entries, as they are typically represented in computers using floating-point numbers. In our subsequent analysis, however, assuming rational entries for just one column of the matrix $\boldsymbol{\Psi}$ is sufficient.

**Lemma 1.** *Let $\boldsymbol{A} \in \mathbb{R}^{D \times T}$, and for one of its rows, like $r$'th row, we have $\boldsymbol{A}_r \in \mathbb{Q}^T$. Then, in every neighborhood of $\boldsymbol{A}$, there is a matrix $\hat{\boldsymbol{A}}$ such that $Ker(\hat{\boldsymbol{A}}) \cap \mathbb{Q}^T = \boldsymbol{O}$.*

---

[1] We opted for the symbol $V$ to represent these points, considering them as cells in a $T$-dimensional von Neumann neighborhood of $K$ from the origin. This clarification is provided to avoid any potential confusion that $V$ denotes a vector space, which is common in mathematical literature.

(The proof is provided in the Appendix C.3.) Consider Lemma equation 1, where we let $\boldsymbol{A} = \boldsymbol{\Psi}^{tr}$. Thus, for every neighborhood of $\boldsymbol{\Psi}^{tr}$, there exists a matrix $\hat{\boldsymbol{\Psi}}^{tr}$ such that $Ker(\hat{\boldsymbol{\Psi}}^{tr}) \cap \mathbb{Q}^T = \boldsymbol{O}$; in other words, $\hat{\boldsymbol{\Psi}}^{tr}$ is injective over rational points, and consequently over integer lattice points. This guarantees that $|\tilde{\mathcal{H}}(\hat{\boldsymbol{\Psi}})| = |V(T, J^{L-1})|$.

In summary, this section demonstrated that, in comparison to SIREN, STAF can substantially increase the size of the set of potential frequencies by a factor of $\tau^K$. This underscores how leveraging the properties of the Kronecker product enables the proposed activation function to significantly enhance expressive power.

## 6 CONCLUSION

In this paper, we introduced STAF as a novel approach to enhancing INRs. Our work mitigates the limitations of conventional ReLU neural networks, particularly their spectral bias which impedes the reconstruction of fine details in target signals. Through experimentation, we demonstrated that STAF significantly outperforms SOTA models like WIRE, SIREN, and Fourier features in terms of accuracy, convergence speed, and PSNR value. Our results demonstrates the effectiveness of STAF in capturing high-frequency details more precisely, which is crucial for applications in computer graphics and data compression. The parametric, trainable nature of STAF allows for adaptive learning tailored to the specific characteristics of the input signals, resulting in superior reconstruction quality. Moreover, our theoretical analysis provided insights into the underlying mechanisms that contribute to the improved performance of STAF. By combining the strengths of Fourier series with the flexibility of neural networks, STAF presents a powerful tool for various high-fidelity signal processing tasks.

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

CONTENTS

## A  NEURAL TANGENT KERNEL

The Neural Tangent Kernel (NTK) is a significant concept in the theoretical understanding of neural networks, particularly in the context of their training dynamics (Jacot et al., 2018). To be self-contained, we provide an explanation of the NTK and its background in kernel methods. We believe this will be beneficial for readers, as previous papers on implicit neural representation using the NTK concept have not adequately explained the NTK or the significance of its eigenvalues and eigenfunctions.

A kernel is a function $K(\mathbf{x}, \tilde{\mathbf{x}})$ used in integral transforms to define an operator that maps a function $f$ to another function $T_f$ through the integral equation

$$T_f(\mathbf{x}) = \int K(\mathbf{x}, \tilde{\mathbf{x}}) f(\tilde{\mathbf{x}}) \, d\tilde{\mathbf{x}}.$$

Since $T_f$ is a linear operator with respect to $f$, we can discuss its eigenvalues and eigenfunctions. The eigenvalues and eigenfunctions of a kernel are the scalar values $\lambda$ and the corresponding functions $\zeta(\mathbf{x})$ that satisfy the following equation (Ghojogh et al., 2021)

$$\int K(\mathbf{x}, \tilde{\mathbf{x}}) \zeta(\tilde{\mathbf{x}}) \, d\tilde{\mathbf{x}} = \lambda \zeta(\mathbf{x}).$$

In the context of neural networks, the concept of a kernel becomes particularly remarkable when analyzing the network's behavior in the infinite-width limit. Kernels in machine learning, such as the Radial Basis Function (RBF) kernel or polynomial kernel, are used to measure similarity between data points in a high-dimensional feature space. These kernels allow the application of linear methods to non-linear problems by implicitly mapping the input data into a higher-dimensional space (Braun, 2005).

The NTK extends this idea by considering the evolution of a neural network's outputs during training. When a neural network is infinitely wide, its behavior can be closely approximated by a kernel method. In this case, the kernel in question is the NTK, which emerges from the first-order Taylor series approximation (or tangent plane approximation) of the network's outputs.

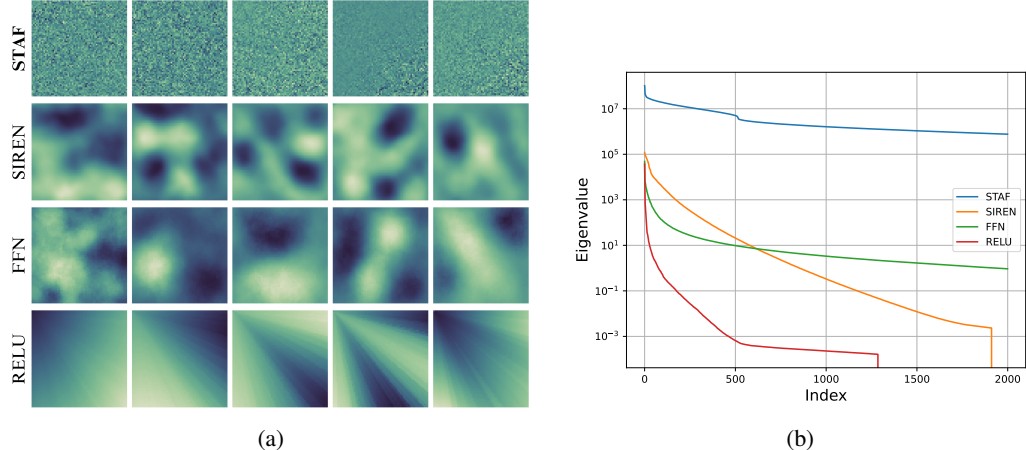

(a)                                                                    (b)

Figure 5: (a) The first five eigenfunctions of the empirical NTK of STAF, SIREN, FFN and ReLU. (b) The eigenvalue spectrum of the empirical NTK of STAF, SIREN, FFN and ReLU.

Formally, for a neural network $f(\mathbf{x}; \boldsymbol{\theta})$ with input $\mathbf{x}$ and parameters $\boldsymbol{\theta}$, the NTK, denoted as $K^{(L)}(\mathbf{x}, \tilde{\mathbf{x}})$, is defined as:

$$K^{(L)}(\mathbf{x}, \tilde{\mathbf{x}}) = \langle \nabla_{\boldsymbol{\theta}} f(\mathbf{x}; \boldsymbol{\theta}), \nabla_{\boldsymbol{\theta}} f(\tilde{\mathbf{x}}; \boldsymbol{\theta}) \rangle,$$

where $\nabla_{\boldsymbol{\theta}} f(\mathbf{x}; \boldsymbol{\theta})$ represents the gradient of the network output with respect to its parameters.

There are two methods for calculating the NTK: the analytic approach and the empirical approach (Novak et al., 2019; Chen et al., 2022). In the paper, we derived the analytic NTK of a neural network that uses our activation function, as detailed in the appendix. However, for our experimental purposes, we utilized the empirical NTK. It is worth noting that calculating the NTK for real-world networks is highly challenging, and typically not computationally possible (Mohamadi et al., 2023).

Just like the computation of NTK, there are analytic and empirical methods for calculating the eigenvalues and eigenfunctions of a kernel (Williams & Seeger, 2000). These values play a crucial role in characterizing neural network training. For instance, it has been shown that the eigenvalues of the NTK determine the convergence rate (Wang et al., 2022; Bai et al., 2023). Specifically, components of the target function associated with kernel eigenvectors having larger eigenvalues are learned faster (Wang et al., 2022; Tancik et al., 2020). In fully-connected networks, the eigenvectors corresponding to higher eigenvalues of the NTK matrix generally represent lower frequency components (Wang et al., 2022). Furthermore, the eigenfunctions of an NTK can illustrate how effectively a model learns a signal dictionary (Yüce et al., 2022).

Figure 5a illustrates the eigenfunctions of various NTKs using different activation functions. As shown, the STAF activation function results in finer eigenfunctions, which intuitively enhances the ability to learn and reconstruct higher frequency components. Additionally, Figure 5b presents the eigenvalues of different NTKs with various activation functions. The results indicate that STAF produces higher eigenvalues, leading to a faster convergence rate during training. Moreover, STAF also generates a greater number of eigenvalues, compared to ReLU and SIREN. Having more eigenvalues is beneficial because it suggests a richer and more expressive kernel, capable of capturing a wider range of features and details in the data.

### A.1 ANALYTIC NTK

In this section, we compute the analytic NTK for a neural network that uses the proposed activation function (STAF), following the notation from (Radhakrishnan, 2024). Interested readers can also refer to (Jacot et al., 2018) and (Golikov et al., 2022). However, we chose (Radhakrishnan, 2024) for its clarity and ease of understanding. According to (Radhakrishnan, 2024), the NTK of an activation function for a neural network with $L - 1$ hidden layers is as follows.

**Theorem 5.** *(Theorem 1 of (Radhakrishnan, 2024), Lecture 6) For $\boldsymbol{x} \in \mathcal{S}^{d-1}$, let $f_{\boldsymbol{x}}^{(L)}(\boldsymbol{w}) : \mathbb{R}^p \to \mathbb{R}$ denote a neural network with $L - 1$ hidden layers such that:*

$$f_{\boldsymbol{x}}^{(L)}(\boldsymbol{w}) = \boldsymbol{W^{(L)}} \frac{1}{\sqrt{F_{L-1}}} \phi \left( \boldsymbol{W^{(L-1)}} \frac{1}{\sqrt{F_{L-2}}} \phi \left( \dots \boldsymbol{W^{(2)}} \frac{1}{\sqrt{F_1}} \phi \left( \boldsymbol{W^{(1)}} \boldsymbol{x} \right) \dots \right) \right); \quad (19)$$

*where $W^{(i)} \in \mathbb{R}^{F_i \times F_{i-1}}$ for $i \in \{1, \dots, L\}$ with $F_0 = d$, $F_L = 1$, and $\phi : \mathbb{R} \to \mathbb{R}$ is an element-wise activation function. As $F_1, F_2, \dots, F_{L-1} \to \infty$ in order, the Neural Network Gaussian Process (NNGP), denoted as $\Sigma^{(L)}$, and the NTK, denoted as $K^{(L)}$, of $f_{\boldsymbol{x}}(\boldsymbol{w})$ are given by:*

$$\Sigma^{(L)}(\boldsymbol{x}, \tilde{\boldsymbol{x}}) = \check{\phi} \left( \Sigma^{(L-1)}(\boldsymbol{x}, \tilde{\boldsymbol{x}}) \right); \quad \Sigma^{(0)}(\boldsymbol{x}, \tilde{\boldsymbol{x}}) = \boldsymbol{x}^T \tilde{\boldsymbol{x}}$$

$$K^{(L)}(\boldsymbol{x}, \tilde{\boldsymbol{x}}) = \Sigma^{(L)}(\boldsymbol{x}, \tilde{\boldsymbol{x}}) + K^{(L-1)}(\boldsymbol{x}, \tilde{\boldsymbol{x}}) \check{\phi}' \left( \Sigma^{(L-1)}(\boldsymbol{x}, \tilde{\boldsymbol{x}}) \right); \quad (20)$$

$$K^{(0)}(\boldsymbol{x}, \tilde{\boldsymbol{x}}) = \boldsymbol{x}^T \tilde{\boldsymbol{x}}$$

*where $\check{\phi} : [-1, 1] \to \mathbb{R}$ is the dual activation for $\phi$, and is calculated as follows:*

$$\check{\phi}(\xi) = \mathbb{E}_{(u,v) \sim \mathcal{N}(0,\boldsymbol{\Lambda})}[\phi(u)\phi(v)] \quad \text{where } \boldsymbol{\Lambda} = \begin{bmatrix} 1 & \xi \\ \xi & 1 \end{bmatrix}. \quad (21)$$

*Furthermore, $\phi$ is normalized such that $\check{\phi}(1) = 1$.*

Consequently, it suffices to calculate $\check{\phi}$. It has been calculated in the following theorem. Just like what mentioned in (Wang et al., 2023), we assume that the optimization of neural networks with STAF can be decomposed into two phases, where we learn the coefficients of STAF in the first phase and then train the parameters of neural network in the second phase. This assumption is reasonable as the number of parameters of STAF is far less than those of networks and they quickly converge at the early stage of training. As a result, in the following theorem, all the parameters except weights are fixed, since they have been obtained in the first phase of training.

**Theorem 6.** *Let $\rho^*$ be the proposed activation function (STAF). Then*

$$\check{\rho}^*(\xi) = \sum_{i=1}^{\tau} \sum_{j=1}^{\tau} C_i C_j \Delta_{i,j}$$

$$= \frac{1}{2} \sum_{i=1}^{\tau} \sum_{j=1}^{\tau} C_i C_j e^{\frac{-1}{2}\left(\Omega_i^2 + \Omega_j^2\right)} \left( e^{\Omega_i \Omega_j \xi} \cos(\Phi_i - \Phi_j) + e^{-\Omega_i \Omega_j \xi} \cos(\Phi_i + \Phi_j) \right) \quad (22)$$

*Therefore,*

$$\check{\rho}^{*\prime}(\xi) = \frac{1}{2} \sum_{i=1}^{\tau} C_i \Omega_i \sum_{j=1}^{\tau} \left[ C_j \Omega_j e^{\frac{-1}{2}\left(\Omega_i^2 + \Omega_j^2\right)} \left( e^{\Omega_i \Omega_j \xi} \cos(\Phi_i - \Phi_j) - e^{-\Omega_i \Omega_j \xi} \cos(\Phi_i + \Phi_j) \right) \right]. \quad (23)$$

*Proof.*

$$\check{\rho}^*(\xi) = \mathbb{E}_{(u,v) \sim \mathcal{N}(0,\boldsymbol{\Lambda})}[\rho^*(u)\rho^*(v)]$$

$$= \mathbb{E}_{(u,v) \sim \mathcal{N}(0,\boldsymbol{\Lambda})} \left[ \sum_{i=1}^{\tau} C_i \sin(\Omega_i u + \Phi_i) \sum_{i=1}^{\tau} C_i \sin(\Omega_i v + \Phi_i) \right]$$

$$= \mathbb{E}_{(u,v) \sim \mathcal{N}(0,\boldsymbol{\Lambda})} \left[ \sum_{i=1}^{\tau} \sum_{j=1}^{\tau} C_i C_j \sin(\Omega_i u + \Phi_i) \sin(\Omega_j v + \Phi_j) \right]$$

$$= \sum_{i=1}^{\tau} \sum_{j=1}^{\tau} C_i C_j \mathbb{E}_{(u,v) \sim \mathcal{N}(0,\boldsymbol{\Lambda})} \left( \sin(\Omega_i u + \Phi_i) \sin(\Omega_j v + \Phi_j) \right). \quad (24)$$

So, we need to compute the following expectation:

$$\Delta_{i,j} = \mathbb{E}_{(u,v) \sim \mathcal{N}(0,\boldsymbol{\Lambda})} \left( \sin(\Omega_i u + \Phi_i) \sin(\Omega_j v + \Phi_j) \right) \quad (25)$$

Note that for a random vector $\mathbf{X} = (X_1, \ldots, X_d)^T$ with mean vector $\boldsymbol{\mu}$ and covariance matrix $\boldsymbol{\Lambda}$, the joint probability density function (PDF) is as follows:

$$f_{\mathbf{X}}(\mathbf{x}) = (2\pi)^{-d/2} \det(\boldsymbol{\Lambda})^{-1/2} e^{(\frac{-1}{2}(\mathbf{x}-\boldsymbol{\mu})^T \boldsymbol{\Lambda}^{-1}(\mathbf{x}-\boldsymbol{\mu}))}. \tag{26}$$

As a result, since $\boldsymbol{\Lambda}^{-1} = \frac{1}{1-\xi^2}\begin{bmatrix} 1 & -\xi \\ -\xi & 1 \end{bmatrix}$, we will have:

$$
\begin{aligned}
f_{U,V}(u,v) &= \frac{1}{2\pi\sqrt{1-\xi^2}} e^{-\frac{1}{2}(u \ \ v)\boldsymbol{\Lambda}^{-1}\begin{pmatrix} u \\ v \end{pmatrix}} = \frac{1}{2\pi\sqrt{1-\xi^2}} e^{\frac{-1}{2(1-\xi^2)}(u \ \ v)\begin{pmatrix} 1 & -\xi \\ -\xi & 1 \end{pmatrix}\begin{pmatrix} u \\ v \end{pmatrix}} \\
&= \frac{1}{2\pi\sqrt{1-\xi^2}} e^{\frac{-(u^2-2\xi uv+v^2)}{2(1-\xi^2)}}.
\end{aligned} \tag{27}
$$

Consequently, using Equations (24) and (25), we have

$$
\begin{aligned}
\Delta_{i,j} &= \int_{-\infty}^{\infty}\int_{-\infty}^{\infty}\Big(\sin(\Omega_i u + \Phi_i)\sin(\Omega_j v + \Phi_j)f_{U,V}(u,v)\Big)dudv \\
&= \frac{1}{2\pi\sqrt{1-\xi^2}}\int_{-\infty}^{\infty}\sin(\Omega_j v + \Phi_j)I_1 dv;
\end{aligned} \tag{28}
$$

where

$$
\begin{aligned}
I_1 &= \int_{-\infty}^{\infty}\sin(\Omega_i u + \Phi_i)e^{\frac{-(u^2-2\xi uv+v^2)}{2(1-\xi^2)}}du = e^{\frac{-v^2}{2(1-\xi^2)}}\int_{-\infty}^{\infty}\sin(\Omega_i u + \Phi_i)e^{\frac{-(u^2-2\xi uv)}{2(1-\xi^2)}}du \\
&= e^{\frac{-v^2+\xi^2 v^2}{2(1-\xi^2)}}\int_{-\infty}^{\infty}\sin(\Omega_i u + \Phi_i)e^{\frac{-(u^2-2\xi uv+\xi^2 v^2)}{2(1-\xi^2)}}du \\
&= e^{-v^2/2}\int_{-\infty}^{\infty}\sin(\Omega_i u + \Phi_i)e^{\frac{-(u-\xi v)^2}{2(1-\xi^2)}}du
\end{aligned} \tag{29}
$$

By assuming $\eta = u - \xi v$ we will have:

$$I_1 = e^{-v^2/2}\int_{-\infty}^{\infty}\sin(\Omega_i(\eta+\xi v)+\Phi_i)e^{\frac{-\eta^2}{2(1-\xi^2)}}d\eta \tag{30}$$

Before going further, we need to consider the following lemma.

**Lemma 2.**

$$\int_{-\infty}^{\infty}\cos(\alpha u + \beta)e^{-\gamma u^2}du = \sqrt{\frac{\pi}{\gamma}}e^{-\frac{\alpha^2}{4\gamma}}\cos\beta, \tag{31}$$

$$\int_{-\infty}^{\infty}\sin(\alpha u + \beta)e^{-\gamma u^2}du = \sqrt{\frac{\pi}{\gamma}}e^{-\frac{\alpha^2}{4\gamma}}\sin\beta \tag{32}$$

*The proof is provided in equation A.2.*

Let $\alpha = \Omega_i$, $\beta = \Omega_i\xi v + \Phi_i$, and $\gamma = \frac{1}{2(1-\xi^2)}$. As a result of equation equation 32, we have

$$
\begin{aligned}
I_1 &= e^{-v^2/2}\sqrt{2\pi(1-\xi^2)}e^{\frac{-\Omega_i^2}{2/(1-\xi^2)}}\sin(\Omega_i\xi v + \Phi_i) \\
&= \sqrt{2\pi(1-\xi^2)}e^{\frac{-(v^2+\Omega_i^2(1-\xi^2))}{2}}\sin(\Omega_i\xi v + \Phi_i)
\end{aligned} \tag{33}
$$

Therefore, based on equation 28, we will have

$$
\begin{aligned}
\Delta_{i,j} &= \frac{1}{2\pi\sqrt{1-\xi^2}}\int_{-\infty}^{\infty}\Big[\sin(\Omega_j v + \Phi_j)\sqrt{2\pi(1-\xi^2)}e^{\frac{-(v^2+\Omega_i^2(1-\xi^2))}{2}}\sin(\Omega_i\xi v + \Phi_i)\Big]dv \\
&= \frac{e^{\frac{(-\Omega_i^2(1-\xi^2))}{2}}}{\sqrt{2\pi}}\int_{-\infty}^{\infty}\Big[\sin(\Omega_j v + \Phi_j)e^{-v^2/2}\sin(\Omega_i\xi v + \Phi_i)\Big]dv \\
&= \frac{e^{-\Omega_i^2(1-\xi^2)/2}}{\sqrt{2\pi}}\int_{-\infty}^{\infty}e^{-v^2/2}\aleph\, dv
\end{aligned} \tag{34}
$$

where

$$\aleph = \frac{1}{2}\Big[\cos(v(\Omega_i\xi - \Omega_j) + \Phi_i - \Phi_j) - \cos(v(\Omega_i\xi + \Omega_j) + \Phi_i + \Phi_j)\Big] \tag{35}$$

Therefore,

$$\Delta_{i,j} = \frac{e^{-\Omega_i^2(1-\xi^2)/2}}{2\sqrt{2\pi}}\left(\sqrt{2\pi}e^{-(\Omega_i\xi-\Omega_j)^2/2}\cos(\Phi_i - \Phi_j) + \sqrt{2\pi}e^{-(\Omega_i\xi+\Omega_j)^2/2}\cos(\Phi_i + \Phi_j)\right)$$

$$= \frac{e^{-\Omega_i^2(1-\xi^2)/2}}{2}\left(e^{-(\Omega_i\xi-\Omega_j)^2/2}\cos(\Phi_i - \Phi_j) + e^{-(\Omega_i\xi+\Omega_j)^2/2}\cos(\Phi_i + \Phi_j)\right)$$

$$= \frac{e^{\frac{-\Omega_i^2(1-\xi^2)}{2}}e^{\frac{-(\Omega_i^2\xi^2+\Omega_j^2)}{2}}}{2}\left(e^{\Omega_i\Omega_j\xi}\cos(\Phi_i - \Phi_j) + e^{-\Omega_i\Omega_j\xi}\cos(\Phi_i + \Phi_j)\right)$$

$$= \frac{e^{\frac{-1}{2}\left(\Omega_i^2+\Omega_j^2\right)}}{2}\left(e^{\Omega_i\Omega_j\xi}\cos(\Phi_i - \Phi_j) + e^{-\Omega_i\Omega_j\xi}\cos(\Phi_i + \Phi_j)\right) \tag{36}$$

As a result of Equations (24) and (36), we have

$$\check{\rho}^*(\xi) = \sum_{i=1}^{\tau}\sum_{j=1}^{\tau}C_iC_j\Delta_{i,j}$$

$$= \frac{1}{2}\sum_{i=1}^{\tau}\sum_{j=1}^{\tau}C_iC_j e^{\frac{-1}{2}\left(\Omega_i^2+\Omega_j^2\right)}\left(e^{\Omega_i\Omega_j\xi}\cos(\Phi_i - \Phi_j) + e^{-\Omega_i\Omega_j\xi}\cos(\Phi_i + \Phi_j)\right) \tag{37}$$

$$\square$$

## A.2 PROOF OF LEMMA EQUATION 2

*Proof.* We want to calculate these integrals:

$$I_1 = \int_{-\infty}^{\infty}\cos(\alpha u + \beta)e^{-\gamma u^2}du,$$

$$I_2 = \int_{-\infty}^{\infty}\sin(\alpha u + \beta)e^{-\gamma u^2}du \tag{38}$$

By adding them we will have

$$I_1 + iI_2 = \int_{-\infty}^{\infty}e^{-\gamma u^2}\big(\cos(\alpha u + \beta) + i\sin(\alpha u + \beta)\big)du = \int_{-\infty}^{\infty}e^{i(\alpha u+\beta)}e^{-\gamma u^2}du$$

$$= e^{i\beta}\int_{-\infty}^{\infty}e^{-\gamma(u^2+\frac{\alpha i}{\gamma}u)}du = e^{i\beta}\int_{-\infty}^{\infty}e^{-\gamma(u^2+\frac{\alpha i}{\gamma}u-\frac{\alpha^2}{4\gamma^2})}e^{-\frac{\alpha^2}{4\gamma}}du$$

$$= e^{-\frac{\alpha^2}{4\gamma}+i\beta}\int_{-\infty}^{\infty}e^{-\gamma(u^2+\frac{\alpha i}{\gamma}u-\frac{\alpha^2}{4\gamma^2})}du = e^{-\frac{\alpha^2}{4\gamma}+i\beta}\underbrace{\int_{-\infty}^{\infty}e^{-\gamma\left(u+\frac{\alpha i}{2\gamma}\right)^2}du}_{I_3} \tag{39}$$

where $i$ is the unit imaginary number. Since we know that the integral of an arbitrary Gaussian function is

$$\int_{-\infty}^{\infty}e^{-a(x+b)^2}dx = \sqrt{\frac{\pi}{a}}, \tag{40}$$

we will have $I_3 = \sqrt{\frac{\pi}{\gamma}}$. Therefore,

$$I_1 + iI_2 = \sqrt{\frac{\pi}{\gamma}}e^{-\frac{\alpha^2}{4\gamma}+i\beta} = \sqrt{\frac{\pi}{\gamma}}e^{-\frac{\alpha^2}{4\gamma}}\left(\cos\beta + i\sin\beta\right) \tag{41}$$

As a result,

$$I_1 = \sqrt{\frac{\pi}{\gamma}}e^{-\frac{\alpha^2}{4\gamma}}\cos\beta, \quad I_2 = \sqrt{\frac{\pi}{\gamma}}e^{-\frac{\alpha^2}{4\gamma}}\sin\beta. \tag{42}$$

$$\square$$

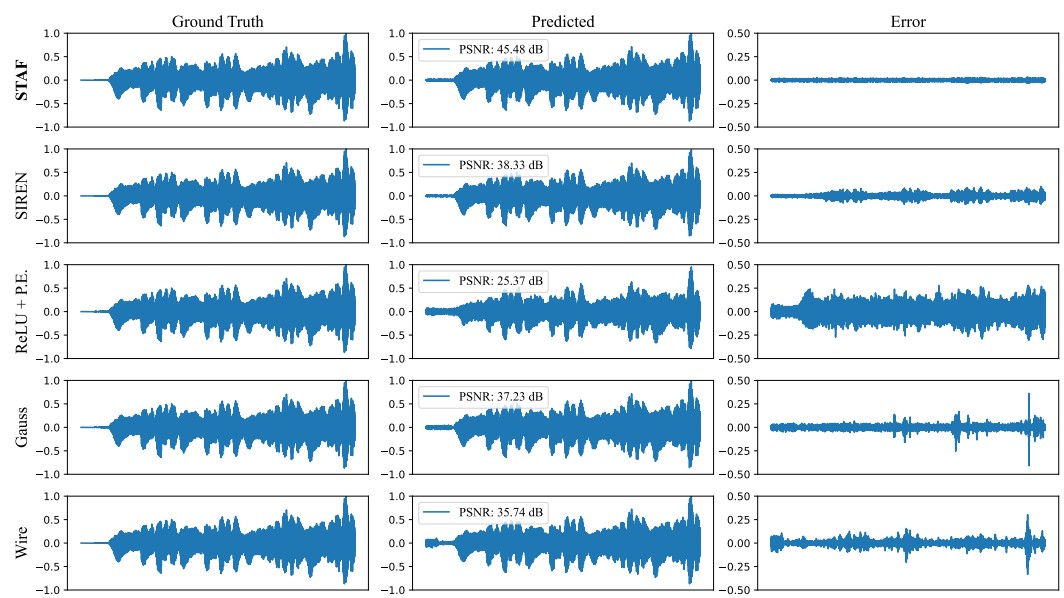

Figure 6: Comparative visualization of audio representation with **STAF** and other activation functions.

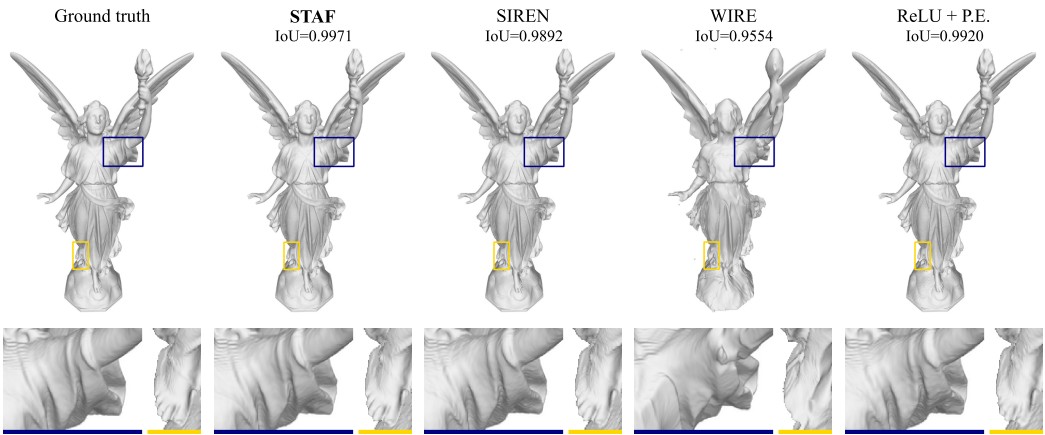

Figure 7: Comparative visualization of shape representation with **STAF** and other activation functions.

## B  ADDITIONAL EXPERIMENTAL RESULTS

In this section, we provide further experimental results to showcase the robustness and efficacy of STAF across different types of data representations. Specifically, we evaluate the performance of STAF in audio and shape representation tasks, comparing it against state-of-the-art activation functions such as SIREN, WIRE, Gaussian, and ReLU with Positional Encoding.

### B.1  AUDIO REPRESENTATION

We used the first 7 seconds of Bach's Cello Suite No. 1: Prelude (Sitzmann et al., 2020), sampled at 44,100 Hz, as our example for the audio representation task. Figure 6 shows the comparative visualization of the audio representation results. The first column presents the ground truth audio waveform, while the second column of each row show the predicted waveforms from each model and their corresponding PSNR values. Additionally, the error plots in the last column highlight areas where each model struggled the most, with brighter regions indicating higher representation errors. STAF achieves the highest PSNR, indicating superior reconstruction fidelity. The SIREN and WIRE

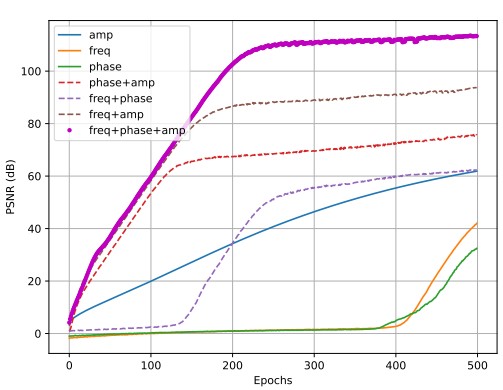

(a) Ablation study of amplitude, frequency, and phase contributions on PSNR performance.

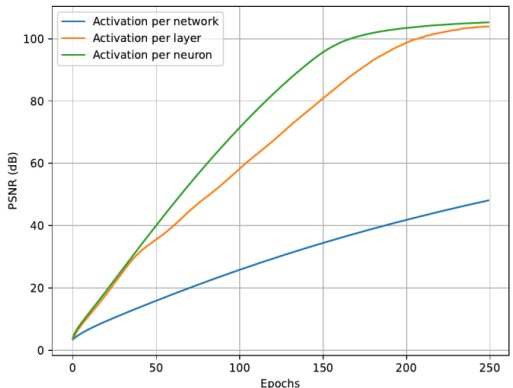

(b) Analysis of activation patterns per network, layer, and neuron on PSNR performance.

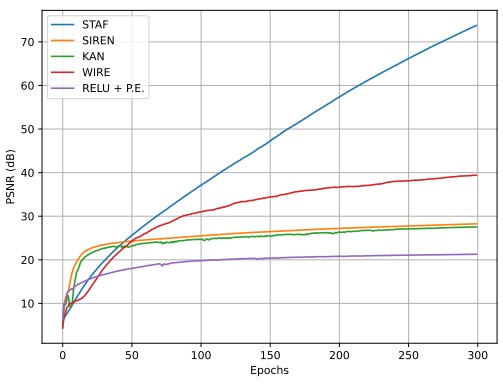

(c) Ablation study on the high-resolution Cameraman image ($256 \times 256$).

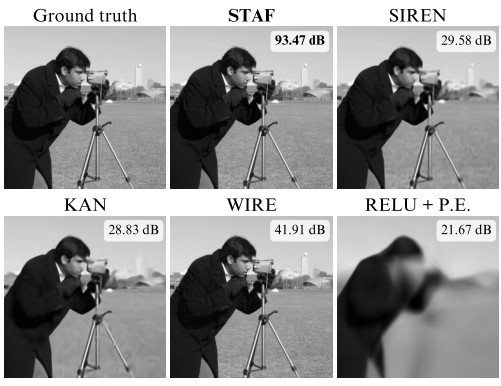

(d) Qualitative comparison of the high-resolution Cameraman image ($256 \times 256$).

Figure 8: Ablation studies exploring various factors influencing model performance and image quality.

models also perform well, but their PSNR values are lower than STAF's, suggesting that STAF can capture finer details in the audio signal.

## B.2 SHAPE REPRESENTATION (OCCUPANCY VOLUME)

We used the Lucy dataset from the Stanford 3D Scanning Repository and followed the WIRE strategy (Saragadam et al., 2023). An occupancy volume was created through point sampling on a 512×512×512 grid, assigning values of 1 to voxels within the object and 0 to voxels outside. Figure 7 illustrates the comparative results for shape representation. The first column displays the ground truth shapes, while the subsequent columns show the reconstructed shapes from each model along with their Intersection over Union (IoU) scores. STAF again demonstrates superior performance with the highest IoU score, closely matching the ground truth shapes. The SIREN and WIRE models show good performance but fall short of STAF's accuracy. The detailed and zoomed plots in second rows of Figure 7 reveal that STAF's reconstructions have fewer discrepancies compared to the other models. This indicates that STAF can better capture complex geometric details, leading to more accurate and high-fidelity shape reconstructions. The enhanced expressive power of STAF, due to its trainable sinusoidal activation functions, allows it to adapt more effectively to the intricacies of 3D shapes.

Overall, the additional experimental results underscore the versatility and effectiveness of STAF across different data representation tasks. By achieving higher PSNR in audio representation and higher IoU in shape representation, STAF proves to be a valuable tool for various applications in computer graphics, audio processing, and beyond.

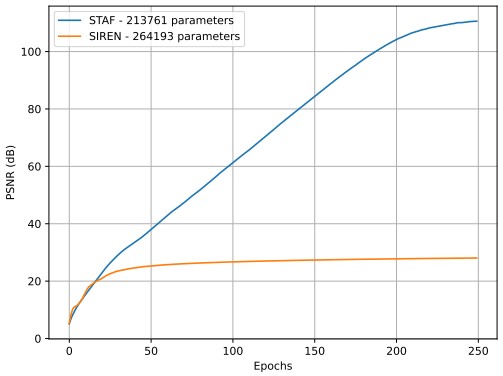
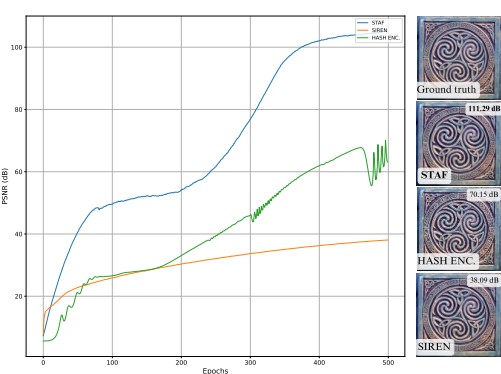

Figure 9: Comparison of PSNR performance between STAF and SIREN over 250 epochs. STAF, with 213,761 parameters, achieves significantly higher PSNR values compared to SIREN, which has 264,193 parameters.

Figure 10: Performance comparison of STAF, SIREN, and Hash Encoding on single image reconstruction. The PSNR curves show that STAF achieves the highest PSNR, followed by Hash Encoding and SIREN.

### B.3 IMPACT OF AMPLITUDE, FREQUENCY, AND PHASE

Figure 8a illustrates the PSNR (dB) over 500 iterations for different component combinations: **amplitude** ($C_i$'s), **frequency** ($\Omega_i$'s), **phase** ($\Phi_i$'s), and their interactions. The model leveraging all three components (freq + phase + amp) achieves the highest PSNR, significantly outperforming individual and partial combinations. This confirms the importance of integrating amplitude, frequency, and phase in the model design for optimal performance, and validates our initial design choices and mathematical analysis. Another observation we derived from this graph is the importance of the parameters. The amplitudes play the most significant role, followed by the frequencies, while the phases are the least important. This insight can be particularly useful when reducing the number of parameters is necessary due to constraints in training time or hardware resources.

### B.4 COMPARATIVE ANALYSIS OF ACTIVATION STRATEGIES

Figure 8b aligns with the described strategies in Section 3.4 for implementing STAF's parametric activation functions. The per-neuron activation (green curve) achieves the highest PSNR, demonstrating superior expressiveness, but at the cost of a significant parameter increase, as expected. The network-wide activation (blue curve) shows the weakest performance, reflecting limited expressiveness due to shared activation functions across the entire network. The layer-wise activation (orange curve) offers a balanced trade-off, achieving nearly the same performance as per-neuron activation while requiring far fewer additional parameters (e.g., 225 parameters for a 3-layer MLP with 25 terms). This supports its use as an efficient and effective strategy, as highlighted in Section 3.4.

### B.5 ABLATION STUDY ON HIGH-RESOLUTION IMAGE RECONSTRUCTION

The ablation study evaluates the performance of various models on a high-resolution `Cameraman` image ($256 \times 256$). The PSNR plot shows that STAF outperforms other models such as SIREN, KAN, WIRE, and ReLU + P.E. across 300 training epochs (Figure 8c). Qualitative results support these findings, with STAF achieving a PSNR of 93.47 dB, outperforming models like KAN (41.91 dB) and WIRE (21.67 dB) at epoch 500 (Figure 8d). These results demonstrate the effectiveness of STAF in high-resolution image reconstruction.

### B.6 PERFORMANCE COMPARISON OF STAF AND SIREN WITH SIMILAR PARAMETER COUNTS

Figure 9 demonstrates the superior performance of STAF compared to SIREN in terms of PSNR (dB) across 250 epochs, despite SIREN having a higher parameter count. To ensure a balanced evaluation, the default configuration of SIREN was modified by adding one additional layer, resulting

in 264,193 parameters for SIREN compared to STAF's 213,761 parameters. This approach avoids extensive parameter tuning for SIREN, offering a practical comparison between the two models. The results clearly show that STAF consistently outperforms SIREN, achieving significantly higher PSNR values throughout the training process. This highlights STAF's efficiency and effectiveness, even when constrained to a lower parameter count, making it a more suitable choice for tasks requiring high-quality image reconstruction.

### B.7 MORE COMPARATIVE EVALUATION

Figure 10 presents a comparative analysis of three methods—STAF, SIREN, and Hash Encoding (Müller et al., 2022) —on the task of high-resolution image reconstruction. The PSNR (dB) curves indicate that STAF significantly outperforms both SIREN and Hash Encoding, reaching a PSNR of over 100 dB after 500 epochs. While Hash Encoding shows a notable improvement over SIREN, peaking at around 70 dB, it still falls short of STAF's superior performance. SIREN, in contrast, exhibits the slowest PSNR growth, achieving only around 38 dB. The qualitative comparisons on the right further support these quantitative results, with STAF closely approximating the ground truth, while Hash Encoding and SIREN produce visibly lower-quality reconstructions. This analysis highlights the advantage of STAF in achieving both higher fidelity and faster convergence in image reconstruction tasks.

## C PROOFS

### C.1 PROOF OF THEOREM EQUATION 1

In this section, we provide a step-by-step proof of Theorem equation 1 concerning the initialization scheme of an architecture that leverages STAF.

**Theorem 7.** *Consider the following function $Z$*

$$Z = \sum_{u=1}^{\tau} C_u \sin\left(\Omega_u \boldsymbol{w}.\boldsymbol{x} + \Phi_u\right) \tag{43}$$

*Suppose $C_u$'s are symmetric distributions, have finite moments, and are independent of $\Omega_u, \boldsymbol{w}, \boldsymbol{x}, \Phi_u$. Furthermore, for each $u$, $\Phi_u \sim U(-\pi, \pi)$. Then the moments of $Z$ will only depend on $\tau$ and the moments of $C_u$'s. Moreover, the odd-order moments of $Z$ will be zero.*

*Proof.* For convenience, let us consider $\Gamma_u = \Omega_u \boldsymbol{w}.\boldsymbol{x}$. Based on the multinomial theorem, for every natural number $q$, we have:

$$Z^q = \sum_{\substack{i_1+\ldots+i_\tau=q \\ i_1,\ldots,i_\tau \geq 0}} \left[ \binom{q}{i_1,\ldots,i_\tau} \prod_{u=1}^{\tau} \left(C_u \sin(\Gamma_u + \Phi_u)\right)^{i_u} \right].$$

According to the linearity of expected value:

$$\mathbb{E}[Z^q] = \sum_{\substack{i_1+\ldots+i_\tau=q \\ i_1,\ldots,i_\tau \geq 0}} \left[ \binom{q}{i_1,\ldots,i_\tau} \mathbb{E}\left[ \prod_{u=1}^{\tau} \left(C_u \sin(\Gamma_u + \Phi_u)\right)^{i_u} \right] \right]$$

$$= \sum_{\substack{i_1+\ldots+i_\tau=q \\ i_1,\ldots,i_\tau \geq 0}} \left[ \binom{q}{i_1,\ldots,i_\tau} \prod_{u=1}^{\tau} \left[ \mathbb{E}[C_u^{i_u}] \mathbb{E}\left[ \sin^{i_u}(\Gamma_u + \Phi_u) \right] \right] \right]. \tag{44}$$

Each choice of $i_1,\ldots,i_\tau$ is called a partition for $q$. If $q$ is an odd number, then in each partition of $q$, at least one of the variables, such as $i_k$, is odd. Since the function $C_i$ is symmetric, it follows that $\mathbb{E}[C_k^{i_k}] = 0$. This is because odd-order moments of a symmetric distribution are always zero. Consequently, the expectation $\mathbb{E}\left[ \prod_{u=1}^{\tau} \left(C_u \sin(\Gamma_u + \Phi_u)\right)^{i_u} \right]$ also equals zero, as does $\mathbb{E}[Z^q]$.

Now, let us consider the case when $q$ is even. For each partition of $q$, if at least one of its variables is odd, then, as before, we have $\mathbb{E}\left[\prod_{u=1}^{\tau}\left(C_u \sin(\Gamma_u + \Phi_u)\right)^{i_u}\right] = 0$. Thus, we can express $q$ as $q = 2j_1 + \ldots + 2j_\tau$ where each $j_k$ is a non-negative integer. According to equation 44, to obtain the $i_k$-th moment of $Z$, we need to calculate $\mathbb{E}\left[\sin^{i_u}(\Gamma_u + \Phi_u)\right]$. In this case, where $i_u = 2j_u$, $\sin^{i_u} \theta$ is an even function, and its Fourier series consists of a constant term and some cosine terms, given by

$$\sin^{2j_u} \theta = \alpha_0 + \sum_{r=1}^{\infty} \alpha_r \cos(r\theta). \tag{45}$$

Hence,

$$\mathbb{E}[\sin^{2j_u}(\Gamma_u + \Phi_u)] = \mathbb{E}[\alpha_0 + \sum_{r=1}^{\infty} \alpha_r \cos(r(\Gamma_u + \Phi_u))] = \alpha_0 + \sum_{r=1}^{\infty} \alpha_r \mathbb{E}[\cos(r\Gamma_u + r\Phi_u)]$$

$$= \alpha_0 + \sum_{r=1}^{\infty} \alpha_r \mathbb{E}[\cos(r\Gamma_u)\cos(r\Phi_u) - \sin(r\Gamma_u)\sin(r\Phi_u)] = \alpha_0 + \sum_{r=1}^{\infty} \alpha_r \Xi \tag{46}$$

where

$$\Xi = \mathbb{E}[\cos(r\Gamma_u)]\mathbb{E}[\cos(r\Phi_u)] - \mathbb{E}[\sin(r\Gamma_u)]\mathbb{E}[\sin(r\Phi_u)]. \tag{47}$$

Since $r$ is an integer, $r\Phi_u$ will be a period, resulting in $\mathbb{E}[\cos(r\Phi_u)] = \mathbb{E}[\sin(r\Phi_u)] = 0$. Thus, $\mathbb{E}[\sin^{2j_u}(\Gamma_u + \Phi_u)] = \alpha_0$.

Using the formula for the coefficients of the Fourier series, we have:

$$\alpha_0 = \frac{1}{\pi} \int_{-\pi/2}^{\pi/2} \sin^{2j_u} \theta \, d\theta = \frac{2}{\pi} \int_0^{\pi/2} \sin^{2j_u} \theta \, d\theta = \frac{2}{\pi} \times \frac{\binom{2j_u}{j_u}}{2^{2j_u}} \times \frac{\pi}{2} = \frac{\binom{2j_u}{j_u}}{2^{2j_u}} \tag{48}$$

where equation 48 is evaluated using the Wallis integral.

To summarize,

$$\mathbb{E}[Z^q] = \sum_{\substack{j_1 + \cdots + j_\tau = \frac{q}{2}, \\ j_1, \ldots, j_\tau \geq 0}} \binom{q}{2j_1, \ldots, 2j_\tau} \prod_{u=1}^{\tau} \mathbb{E}[C_u^{2j_u}] \frac{\binom{2j_u}{j_u}}{2^{2j_u}}$$

$$= \sum_{\substack{j_1 + \cdots + j_\tau = \frac{q}{2}, \\ j_1, \ldots, j_\tau \geq 0}} \left[\left(\binom{q}{2j_1, \ldots, 2j_\tau} \prod_{u=1}^{\tau} \binom{2j_u}{j_u}\right) \prod_{u=1}^{\tau} \frac{1}{2^{2j_u}} \prod_{u=1}^{\tau} \mathbb{E}[C_u^{2j_u}]\right] \tag{49}$$

This also accounts for odd-order moments, as it is impossible to select a combination of non-negative integers that sums to a non-integer value.

It is worth noting that:

$$\binom{q}{2j_1, \ldots, 2j_\tau} \prod_{u=1}^{\tau} \binom{2j_u}{j_u} = \frac{q!}{(2j_1)! \ldots (2j_\tau)!} \times \frac{(2j_1)!}{(j_1)!^2} \times \cdots \times \frac{(2j_\tau)!}{(j_\tau)!^2} = \frac{q!}{(j_1!)^2 \ldots (j_\tau!)^2}$$

$$= \binom{q}{j_1, j_1, \ldots, j_\tau, j_\tau} \tag{50}$$

Furthermore,

$$\prod_{u=1}^{\tau} \frac{1}{2^{2j_u}} = \frac{1}{2^{2\sum_{u=1}^{\tau} j_u}} = \frac{1}{2^q} \tag{51}$$

By utilizing Equations equation 49 to equation 51, we can conclude that:

$$\mathbb{E}[Z^q] = \frac{1}{2^q} \sum_{\substack{j_1 + \cdots + j_\tau = \frac{q}{2} \\ j_1, \ldots, j_\tau \geq 0}} \binom{q}{j_1, j_1, \ldots, j_\tau, j_\tau} \prod_{u=1}^{\tau} \mathbb{E}[C_u^{2j_u}] \tag{52}$$

As you can see, the moments of $Z$ depend solely on $\tau$ and the moments of the $C_u$'s. $\qquad \square$

Now, our goal is to determine the distribution of the $C_u$'s so that the distribution of $Z$ becomes $\mathcal{N}(0,1)$. To achieve this, let's first consider the following theorem:

**Theorem 8.** *(Page 353 of (Shiryaev, 2016)) Let $X \sim \mathcal{N}(0, \sigma^2)$. Then*

$$E(X^q) = \begin{cases} 0, & \text{if } q \text{ is odd} \\ \frac{q!}{\frac{q}{2}! \, 2^{q/2}} \sigma^q, & \text{if } q \text{ is even} \end{cases} \tag{53}$$

*and these moments pertain exclusively to the normal distribution.*

In theorem equation 7, we proved that for odd values of $q$, $\mathbb{E}[h^q] = 0$. Thus, in order to have $Z \sim \mathcal{N}(0,1)$, for even values of $q$, we must have $\mathbb{E}[h^q] = \frac{q!}{\frac{q}{2}! \, 2^{q/2}}$. Alternatively, we can express it as

$$\frac{1}{2^q} \sum_{\substack{j_1 + \cdots + j_\tau = \frac{q}{2} \\ j_1, \ldots, j_\tau \geq 0}} \binom{q}{j_1, j_1, \ldots, j_\tau, j_\tau} \prod_{u=1}^{\tau} \mathbb{E}[C_u^{2j_u}] = \frac{q!}{\frac{q}{2}! \, 2^{q/2}}. \tag{54}$$

Simplifying further, we obtain

$$\frac{q!}{2^q} \sum_{\substack{j_1 + \cdots + j_\tau = \frac{q}{2} \\ j_1, \ldots, j_\tau \geq 0}} \frac{\prod_{u=1}^{\tau} \mathbb{E}[C_u^{2j_u}]}{(j_1!)^2 \cdots (j_\tau!)^2} = \frac{q!}{\frac{q}{2}! \, 2^{q/2}}. \tag{55}$$

This equation can be further simplified to

$$\sum_{\substack{j_1 + \cdots + j_\tau = \frac{q}{2} \\ j_1, \ldots, j_\tau \geq 0}} \frac{\prod_{u=1}^{\tau} \mathbb{E}[C_u^{2j_u}]}{(j_1!)^2 \cdots (j_\tau!)^2} = \frac{2^{q/2}}{\frac{q}{2}!}. \tag{56}$$

Equation equation 56 provides a general formula that can be utilized in further research. It allows for finding different solutions for $C_u$ under various assumptions (e.g., independence or specific dependencies) and different values of $\tau$. However, in the subsequent analysis, we assume that $C_u$'s are independent and identically distributed (i.i.d) random variables. The following theorem aims to satisfy Equation equation 56.

**Theorem 9.** *Suppose $C_u$'s are i.i.d random variables with the following even-order moments:*

$$\mathbb{E}[C_u^{2j}] = \left(\frac{2}{\tau}\right)^j j! \tag{57}$$

*Then, for every non-negative even number $q$, Equation equation 56 holds.*[2]

*Proof.* We begin by simplifying the expression:

$$\sum_{\substack{j_1 + \cdots + j_\tau = \frac{q}{2} \\ j_1, \ldots, j_\tau \geq 0}} \frac{\prod_{u=1}^{\tau} \mathbb{E}[C_u^{2j_u}]}{(j_1!)^2 \cdots (j_\tau!)^2} = \sum_{\substack{j_1 + \cdots + j_\tau = \frac{q}{2} \\ j_1, \ldots, j_\tau \geq 0}} \frac{\prod_{u=1}^{\tau} \left[\left(\frac{2}{\tau}\right)^j j!\right]}{(j_1!)^2 \cdots (j_\tau!)^2}$$

$$= \sum_{\substack{j_1 + \cdots + j_\tau = \frac{q}{2} \\ j_1, \ldots, j_\tau \geq 0}} \left(\frac{2}{\tau}\right)^{\sum_{u=1}^{\tau} j_u} \left(\frac{1}{j_1! \ldots j_\tau!}\right) = \sum_{\substack{j_1 + \cdots + j_\tau = \frac{q}{2} \\ j_1, \ldots, j_\tau \geq 0}} \left(\frac{2}{\tau}\right)^{\frac{q}{2}} \left(\frac{1}{j_1! \ldots j_\tau!}\right)$$

$$= \left(\frac{2}{\tau}\right)^{\frac{q}{2}} \sum_{\substack{j_1 + \cdots + j_\tau = \frac{q}{2} \\ j_1, \ldots, j_\tau \geq 0}} \frac{1}{j_1! \ldots j_\tau!} = \left(\frac{2}{\tau}\right)^{\frac{q}{2}} \frac{1}{(\frac{q}{2})!} \sum_{\substack{j_1 + \cdots + j_\tau = \frac{q}{2} \\ j_1, \ldots, j_\tau \geq 0}} \frac{(\frac{q}{2})!}{j_1! \ldots j_\tau!}$$

$$= \left(\frac{2}{\tau}\right)^{\frac{q}{2}} \frac{1}{(\frac{q}{2})!} \sum_{\substack{j_1 + \cdots + j_\tau = \frac{q}{2} \\ j_1, \ldots, j_\tau \geq 0}} \binom{\frac{q}{2}}{j_1, \ldots, j_\tau} \tag{58}$$

---

[2]If you wonder how this solution struck our mind, you can start by solving equation equation 56 for $q = 2$ to obtain $\mathbb{E}[h^2]$. Then, using the value of $\mathbb{E}[h^2]$, solve equation equation 56 for $q = 4$ to obtain $\mathbb{E}[h^4]$, and so on.

Based on the multinomial theorem, we can conclude that

$$\left(\frac{2}{\tau}\right)^{\frac{q}{2}} \frac{1}{(\frac{q}{2})!} \sum_{\substack{j_1+\cdots+j_\tau=\frac{q}{2} \\ j_1,\ldots,j_\tau \geq 0}} \binom{\frac{q}{2}}{j_1,\ldots,j_\tau} = \left(\frac{2}{\tau}\right)^{\frac{q}{2}} \frac{\tau^{\frac{q}{2}}}{(\frac{q}{2})!} = \frac{2^{\frac{q}{2}}}{(\frac{q}{2})!} \tag{59}$$

$\square$

Also note that according to Theorem equation 7, the odd-order moments of $Z$ are zero, just like a normal distribution.

**Corollary 1.** *Let $Z$ be the random variable defined in equation 43. Additionally, assume that the $C_u$'s ($1 \leq u \leq \tau$) used in the definition of $Z$, are i.i.d random variables with even moments as defined in theorem equation 9. Then $Z \sim \mathcal{N}(0,1)$.*

*Proof.* We know that if the MGF of a distribution exists, then the moments of that distribution can uniquely determine its PDF. That is, if $X$ and $Y$ are two distributions and for every natural number $k$, $E(X^k) = E(Y^k)$, then $X = Y$.

In the Theorem equation 9, we observed that the moments of $Z$ are equal to the moments of a standard normal distribution. Since the MGF of this distribution exists, $Z \sim \mathcal{N}(0,1)$. $\square$

Now, let's explore which distribution can produce the moments defined in equation equation 57. To have an inspiration, note that for a centered Laplace random variable $X$ with scale parameter $b$, we have the PDF of $X$ as

$$f_X(x) = \frac{1}{2b} e^{\frac{-|x|}{b}} \tag{60}$$

and the moments of $X$ given by

$$\mathbb{E}[X^q] = \begin{cases} 0, & \text{if } q \text{ is odd} \\ \frac{b^q}{q!}, & \text{if } q \text{ is even} \end{cases} \tag{61}$$

Hence, the answer might be similar to this distribution. If we assume $Y = sgn(X)\sqrt{|X|}$, since $Y$ is symmetric, all of its odd-order moments are zero. Now, let us calculate its even-order moments:

$$\mathbb{E}[Y^{2q}] = \mathbb{E}[|X|^q] = \int_{-\infty}^{\infty} |x|^q \frac{1}{2b} e^{-\frac{|x|}{b}} dx = 2 \int_0^{\infty} |x|^q \frac{1}{2b} e^{-\frac{|x|}{b}} dx = \frac{1}{b} \int_0^{\infty} x^q e^{-\frac{x}{b}} dx \tag{62}$$

By assuming $u = \frac{x}{b}$, we will have

$$\mathbb{E}[Y^{2q}] = \int_0^{\infty} (bu)^q e^{-u} du = b^q \int_0^{\infty} u^q e^{-u} du = b^q \Gamma(q+1) = b^q q! \tag{63}$$

By assuming $b = \frac{2}{\tau}$, equation 57 will be obtained.

The next theorem will obtain the probability distribution function of $Y$.

**Theorem 10.** *Let $X$ be a centered Laplace random variable with scale parameter $b$, and $Y = sgn(X)\sqrt{|X|}$. Then*

$$f_Y(y) = \frac{|y|}{b} e^{\frac{-y^2}{b}} \tag{64}$$

*Proof.* Let $A = Y^2 = |X|$. Therefore,

$$M_A(t) = \sum_{k=0}^{\infty} \frac{t^k \mathbb{E}[|X|^k]}{k!} \tag{65}$$

As we calculated in equation 63, $\mathbb{E}[|X|^k] = b^k k!$. Therefore,

$$M_A(t) = \sum_{k=0}^{\infty} \frac{t^k \cdot b^k k}{k!} = \sum_{k=0}^{\infty} (bt)^k = \frac{1}{1-bt} = \frac{\frac{1}{b}}{\frac{1}{b}-t} \tag{66}$$

that is the MGF of exponential distribution with parameter $\frac{1}{b}$. That is,

$$f_A(a) = \frac{1}{b}e^{\frac{-a}{b}} \tag{67}$$

Therefore, using the fact that $A$ is a always non-negative, we consider non-negative values $a^2$ to describe its cumulative distribution function.

$$F_A(y^2) = \mathbb{P}(A \leq y^2) = 1 - e^{\frac{-y^2}{b}} \tag{68}$$

On the other hand, if $y \geq 0$,

$$\mathbb{P}(A \leq y^2) = \mathbb{P}(Y^2 \leq y^2) = \mathbb{P}(-y \leq Y \leq y) \tag{69}$$

Since we want $Y$ to be symmetric, we assume[3]

$$\mathbb{P}(-y \leq Y \leq y) = 2\,\mathbb{P}(0 \leq Y \leq y) = 2\left(\mathbb{P}(Y \leq y) - \frac{1}{2}\right) = 2F_Y(y) - 1, \quad y \geq 0 \tag{70}$$

Using equations equation 68 to equation 70, we draw conclusion that

$$2F_Y(y) - 1 = 1 - e^{\frac{-y^2}{b}}, \quad y \geq 0 \tag{71}$$

By differentiating both sides of equation 71 with respect to $y$, we will have

$$2f_Y(y) = \frac{2y}{b}e^{\frac{-y^2}{b}}, \quad y \geq 0 \tag{72}$$

Therefore,

$$f_Y(y) = \frac{y}{b}e^{\frac{-y^2}{b}}, \quad y \geq 0 \tag{73}$$

Since we assumed $y \geq 0$ in the above equations, and we supposed that $Y$ is symmetric,

$$f_Y(y) = \frac{|y|}{b}e^{\frac{-y^2}{b}}, \quad y \in \mathbb{R} \tag{74}$$

Just to make sure that our assumption about the symmetry of $Y$ was correct (or sufficed for our purpose), let us check the even-order moments of $Y$. The odd-orders are zero based on the symmetry.

$$\mathbb{E}[Y^{2k}] = \int_{-\infty}^{\infty} y^{2k}\left(\frac{|y|}{b}e^{-\frac{y^2}{b}}\right)dy = \frac{2}{b}\int_0^{\infty} y^{2k+1}e^{-\frac{y^2}{b}}dy \tag{75}$$

Setting $y^2 = t$ and $\frac{1}{b} = s$, leads to the following equation:

$$\mathbb{E}[Y^{2k}] = \frac{1}{b}\int_0^{\infty} t^k e^{-st}dt \tag{76}$$

That is the Laplace transform of $t^k$. Therefore,

$$\mathbb{E}[Y^{2k}] = s\frac{\Gamma(k+1)}{s^{k+1}} = \frac{k!}{s^k} = b^k k! \tag{77}$$

$\square$

In summary, in this section we calculated the initial coefficients of our activation function as described in Theorem equation 10, where we set $b = \frac{2}{\tau}$. Consequently, if we denote the post-activation of layer $l$ by $\mathbf{z}^{(l)}$, we will have $z_i^{(l)} \sim \mathcal{N}(0, 1)$ for all $l \in \{2, 3, \ldots, L-1\}$, and $i \in \{1, \ldots F_l\}$. This result can be proved by induction on $l$, using the fact that, based on the theorems in this section, the PDF of $Z$ is independent of the PDF of $\mathbf{x}$.

---

[3]In fact, the assumption that $Y$ is symmetric is not unexpected, since all odd-order moments of $Y$ are zero. But there are some non-symmetrical distributions whose all odd-order moments are zero (Churchill, 1946). Nevertheless, under some assumptions, it can be shown that a distribution is symmetric if and only if all its odd-order moments are zero. However, we don't use this claim in this paper.

## C.2 Proof of Theorem equation 3

Before proving the theorem, note the following remark:

**Remark 5.** *Let $X$ be a $\chi_1 \times \chi_2$ matrix, and $Y$ be a $\gamma_1 \times \gamma_2$ matrix. Then, according to (Ashendorf et al., 2014; Albrecht et al., 2023):*

$$(X \otimes Y)_{i,j} = x_{\lceil i/\gamma_1 \rceil,\ \lceil i/\gamma_2 \rceil}\ y_{(i-1)\%\gamma_1+1,\ (j-1)\%\gamma_2+1}. \tag{78}$$

Now, let us consider each pair of layers as a block, where the first two layers form the first block, the second two layers form the second block, and so on. We prove the theorem by induction on the block numbers. The proof consists of three parts:

**Part 1)** Consider the weight matrix and bias vector given by:

$$\overline{W^{(l)}} = \Omega \otimes W^{(l)}, \qquad \overline{B^{(l)}} = \Phi \otimes J_{F_l,1}. \tag{79}$$

We then define

$$\left[ \overline{a_1^{(l)}} \quad \overline{a_2^{(l)}} \quad \ldots \quad \overline{a_{\tau F_l}^{(l)}} \right]^{tr} = \overline{W^{(l)}} z^{(l-1)} + \overline{B^{(l)}}, \tag{80}$$

and

$$\overline{z_p^{(l)}} = \rho(\overline{a_p^{(l)}}) \quad \forall\, p \in \{1, 2, \ldots, \tau F_l\}. \tag{81}$$

Additionally, define

$$\tilde{a}^{(l+1)} = \left( C^{tr} \otimes W_{i,:}^{(l+1)} \right) \overline{z^{(l)}}, \tag{82}$$

where $W_{i,:}^{(l+1)}$ denotes the $i$'th row of $W^{(l+1)}$. Then, we can observe that

$$\tilde{a}^{(l+1)} = a_i^{(l+1)} \tag{83}$$

*Proof.* First, let us calculate $a_i^{(l+1)}$ using activation function $\rho^*$. Note that $a^{(l+1)} = W^{(l+1)} z^{(l)}$. Therefore, $a_i^{(l+1)} = W_{i,:}^{(l+1)} z^{(l)}$. It implies that

$$a_i^{(l+1)} = \sum_{j=1}^{F_l} W_{i,j}^{(l+1)} z_j^{(l)} = \sum_{j=1}^{F_l} W_{i,j}^{(l+1)} \rho^* \left( a_j^{(l)} \right) = \sum_{j=1}^{F_l} W_{i,j}^{(l+1)} \rho^* \left( \sum_{k=1}^{F_{l-1}} W_{j,k}^{(l)} z_k^{(l-1)} \right)$$

$$= \sum_{j=1}^{F_l} W_{i,j}^{(l+1)} \sum_{m=1}^{\tau} C_m \rho \left( \Omega_m \sum_{k=1}^{F_{l-1}} W_{j,k}^{(l)} z_k^{(l-1)} + \Phi_m \right) \tag{84}$$

Next, let us calculate $\tilde{a}^{(l+1)}$. We have

$$\overline{a_p^{(l)}} = \left[ \overline{W^{(l)}} z^{(l-1)} + \overline{B^{(l)}} \right]_p = \overline{W^{(l)}}_{p,:} z^{(l-1)} + \overline{B^{(l)}}_p = \sum_{k=1}^{F_{l-1}} \left( \overline{W^{(l)}}_{p,k} z_k^{(l-1)} \right) + \overline{B^{(l)}}_p$$

$$= \sum_{k=1}^{F_{l-1}} \left( \Omega_{\lceil p/F_l \rceil, \lceil k/F_{l-1} \rceil} W_{1+(p-1)\%F_l,\ 1+(k-1)\%F_{l-1}}^{(l)} z_k^{(l-1)} \right) + \Phi_{\lceil p/F_l \rceil} \tag{85}$$

Equation equation 85 is based on equation equation 78. Since $1 \le k \le F_{l-1}$, it follows that $\lceil k/F_{l-1} \rceil = 1$ and $(k-1)\%F_{l-1} = k-1$. As a result,

$$\overline{a_p^{(l)}} = \sum_{k=1}^{F_{l-1}} \left( \Omega_{\lceil p/F_l \rceil} W_{1+(p-1)\%F_l,\ k}^{(l)} z_k^{(l-1)} \right) + \Phi_{\lceil p/F_l \rceil}$$

$$= \Omega_{\lceil p/F_l \rceil} \sum_{k=1}^{F_{l-1}} \left( W_{1+(p-1)\%F_l,\ k}^{(l)} z_k^{(l-1)} \right) + \Phi_{\lceil p/F_l \rceil} \tag{86}$$

Therefore,

$$\overline{z_p^{(l)}} = \rho \left( \mathbf{\Omega}_{\lceil p/F_l \rceil} \sum_{k=1}^{F_{l-1}} \left( W_{1+(p-1)\%F_l,k}^{(l)} z_k^{(l-1)} \right) + \mathbf{\Phi}_{\lceil p/F_l \rceil} \right) \tag{87}$$

Consequently,

$$\tilde{a}^{(l+1)} = \sum_{p=1}^{\tau F_l} \left[ \mathbf{C}^{tr} \otimes W_{i,:}^{(l+1)} \right]_{1,p} \overline{z_p^{(l)}} = \sum_{p=1}^{\tau F_l} \mathbf{C}_{1,\lceil p/F_l \rceil}^{tr} W_{i,1+(p-1)\%F_l}^{(l+1)} \overline{z_p^{(l)}}$$

$$= \sum_{p=1}^{\tau F_l} \mathbf{C}_{\lceil p/F_l \rceil} W_{i,1+(p-1)\%F_l}^{(l+1)} \overline{z_p^{(l)}} \tag{88}$$

$$= \sum_{j=1}^{F_l} \sum_{m=1}^{\tau} \mathbf{W}_{i,j}^{(l+1)} \mathbf{C}_m \overline{z_{F_l(m-1)+j}^{(1)}} \tag{89}$$

Equation equation 89 is obtained as follows: by changing the indices of $\mathbf{W}$ and $\mathbf{C}$ from equation equation 88 to equation 89, we need to change the index of $z^{(l)}$ too. To this end, note that

$$m = \lceil p/F_l \rceil, \quad j = 1 + (p-1)\%F_l \tag{90}$$

If $F_l \nmid p$, then $m = 1 + \lfloor p/F_l \rfloor$. As we know, $p = F_l \lfloor p/F_l \rfloor + p\%F_l$. Therefore, $p = F_l(m-1) + j$. This equation also holds when $F_l \mid p$.

Equation equation 89 can be rewritten as follows:

$$\sum_{j=1}^{F_l} \mathbf{W}_{i,j}^{(l+1)} \sum_{m=1}^{\tau} \mathbf{C}_m \overline{z_{F_l(m-1)+j}^{(l)}} \tag{91}$$

where, according to equations equation 87 and equation 90,

$$\overline{z_{F_l(m-1)+j}^{(l)}} = \rho \left( \mathbf{\Omega}_m \sum_{k=1}^{F_{l-1}} \left( W_{j,k}^{(l)} z_k^{(l-1)} \right) + \mathbf{\Phi}_m \right) \tag{92}$$

Hence,

$$\tilde{a}^{(l+1)} = \sum_{j=1}^{F_l} \mathbf{W}_{i,j}^{(l+1)} \sum_{m=1}^{\tau} \mathbf{C}_m \rho \left( \mathbf{\Omega}_m \sum_{k=1}^{F_{l-1}} \left( W_{j,k}^{(l)} z_k^{(l-1)} \right) + \mathbf{\Phi}_m \right) \tag{93}$$

which is equal to $a_i^{(l+1)}$ based on equation 84. $\qquad\qquad\square$

**Part 2)** Let $\overline{\mathbf{B}^{(l+1)}} = \mathbf{\Phi} \otimes \mathbf{J}_{F_{l+1},1}$. We can define $\overline{a^{(l+1)}}$ as follows:

$$\left[ \overline{a_1^{(l+1)}} \quad \overline{a_2^{(l+1)}} \quad \dots \quad \overline{a_{\tau(F_{l+1})}^{(l+1)}} \right]^{tr} = \mathbf{\Omega} \otimes \boldsymbol{a}^{(l+1)} + \overline{\mathbf{B}^{(l+1)}}. \tag{94}$$

Therefore, using Equations (82), (83) and (94), we can write

$$\overline{a^{(l+1)}} = \overline{W^{(l+1)}} \, \overline{z^{(l)}} + \overline{\mathbf{B}^{(l+1)}} \tag{95}$$

, where

$$\overline{W^{(l+1)}} = \mathbf{\Omega} \otimes \left( \mathbf{C}^{tr} \otimes W^{(l+1)} \right) = \left( \mathbf{\Omega} \otimes \mathbf{C}^{tr} \right) \otimes W^{(l+1)}. \tag{96}$$

Moreover, if we define

$$\overline{z_q^{(l+1)}} = \rho \left( \overline{a_q^{(l+1)}} \right) \quad \forall q \in \{1, \dots, \tau(F_{l+1})\}, \tag{97}$$

we can observe that

$$\boldsymbol{z^{(l+1)}} = \left( \mathbf{C}^{tr} \otimes \mathbf{I}_{F_{l+1}} \right) \overline{\boldsymbol{z^{(l+1)}}}. \tag{98}$$

*Proof.* We know that

$$z_i^{(l+1)} = \rho^*(a_i^{(l+1)}) = \sum_{n=1}^{\tau} \rho\left(\mathbf{\Omega}_n a_i^{(l+1)} + \mathbf{\Phi}_n\right). \tag{99}$$

Now, let us calculate each entry of the RHS of Equation equation 98

$$\left[\left(\mathbf{C}^{tr} \otimes \mathbf{I}_{F_{l+1}}\right)\overline{\mathbf{z}^{(l+1)}}\right]_i = \left[\mathbf{C}^{tr} \otimes \mathbf{I}_{F_{l+1}}\right]_i \overline{\mathbf{z}^{(l+1)}} = \sum_{j=1}^{\tau F_{l+1}} \left(\mathbf{C}^{tr} \otimes \mathbf{I}_{F_{l+1}}\right)_{i,j} \overline{z_j^{(l+1)}}. \tag{100}$$

Hence, according to equation 78, we have

$$\left[\left(\mathbf{C}^{tr} \otimes \mathbf{I}_{F_{l+1}}\right)\overline{\mathbf{z}^{(l+1)}}\right]_i = \sum_{j=1}^{\tau F_{l+1}} \mathbf{C}^{tr}_{\lceil i/F_{l+1}\rceil,\lceil j/F_{l+1}\rceil} \delta_{1+(i-1)\%F_{l+1},1+(j-1)\%F_{l+1}} \overline{z_j^{(l+1)}}, \tag{101}$$

in which $\delta$ refers to Kronecker delta. As a result,

$$\left[\left(\mathbf{C}^{tr} \otimes \mathbf{I}_{F_{l+1}}\right)\overline{\mathbf{z}^{(l+1)}}\right]_i = \sum_{j=1}^{\tau F_{l+1}} \mathbf{C}_{\lceil j/F_{l+1}\rceil,\lceil i/F_{l+1}\rceil} \delta_{1+(i-1)\%F_{l+1},1+(j-1)\%F_{l+1}} \overline{z_j^{(l+1)}} \tag{102}$$

Note that $1 \le i \le F_{l+1}$. Therefore, $\lceil i/F_{l+1}\rceil = 1$, and $(i-1)\%F_{l+1} = i-1$. Hence,

$$\left[\left(\mathbf{C}^{tr} \otimes \mathbf{I}_{F_{l+1}}\right)\overline{\mathbf{z}^{(l+1)}}\right]_i = \sum_{j=1}^{\tau F_{l+1}} \mathbf{C}_{\lceil j/F_{l+1}\rceil} \delta_{i,1+(j-1)\%F_{l+1}} \overline{z_j^{(l+1)}}. \tag{103}$$

Also note that $\delta_{i,1+(j-1)\%F_{l+1}}$ is zero, except when $j = kF_{l+1}+i$, in which case $\delta_{i,1+(j-1)\%F_{l+1}} = 1$. Thus,

$$\left[\left(\mathbf{C}^{tr} \otimes \mathbf{I}_{F_{l+1}}\right)\overline{\mathbf{z}^{(l+1)}}\right]_i = \sum_{k=0}^{\tau-1} \mathbf{C}_{\lceil (kF_{l+1}+i)/F_{l+1}\rceil} \overline{z_{kF_{l+1}+i}^{(l+1)}} = \sum_{k=0}^{\tau-1} \mathbf{C}_{k+\lceil i/F_{l+1}\rceil} \overline{z_{kF_{l+1}+i}^{(l+1)}}$$

$$= \sum_{k=0}^{\tau-1} \mathbf{C}_{k+1} \overline{z_{kF_{l+1}+i}^{(l+1)}} = \sum_{n=1}^{\tau} \mathbf{C}_n \overline{z_{(n-1)F_{l+1}+i}^{(l+1)}} = \sum_{n=1}^{\tau} \mathbf{C}_n \rho\left(\overline{a_{(n-1)F_{l+1}+i}^{(l+1)}}\right). \tag{104}$$

Note that

$$\overline{a_{(n-1)F_{l+1}+i}^{(l+1)}} = \mathbf{\Omega}_{\lceil ((n-1)F_{l+1}+i)/F_{l+1}\rceil} a_{1+((n-1)F_{l+1}+i-1)\%F_{l+1}}^{(l+1)} + \mathbf{\Phi}_{\lceil ((n-1)F_{l+1}+i)/F_{l+1}\rceil}$$

$$= \mathbf{\Omega}_{n-1+\lceil i/F_{l+1}\rceil} a_{1+(i-1)\%F_{l+1}}^{(l+1)} + \mathbf{\Phi}_{n-1+\lceil i/F_{l+1}\rceil} \tag{105}$$

Since $\left\lceil \frac{i}{F_{l+1}}\right\rceil = 1$ and $(i-1)\%F_{l+1} = i-1$, we have

$$\overline{a_{(n-1)F_{l+1}+i}^{(l+1)}} = \mathbf{\Omega}_n a_i^{(l+1)} + \mathbf{\Phi}_n \tag{106}$$

Finally, utilizing Equations equation 104 and equation 106, we deduce that

$$\left[\left(\mathbf{C}^{tr} \otimes \mathbf{I}_{F_{l+1}}\right)\overline{\mathbf{z}^{(l+1)}}\right]_i = \sum_{n=1}^{\tau} \mathbf{C}_n \rho\left(\mathbf{\Omega}_n a_i^{(l+1)} + \mathbf{\Phi}_n\right), \tag{107}$$

which is equal to the RHS of the Equation equation 98.

**Part 3)** Using parts 1 and 2 of the proof, we can state the theorem for arbitrary even values of $L$. By setting $l = 1$ in the previous parts, we obtain

$$\overline{\mathbf{W}^{(1)}} = \mathbf{\Omega} \otimes \mathbf{W}^{(1)}, \quad \overline{\mathbf{B}^{(1)}} = \mathbf{\Phi} \otimes \mathbf{J}_{F_1,1} \tag{108}$$

and

$$\overline{\mathbf{W}^{(2)}} = \left(\mathbf{\Omega} \otimes \mathbf{C}^{tr}\right) \otimes \mathbf{W}^{(2)}, \quad \overline{\mathbf{B}^{(2)}} = \mathbf{\Phi} \otimes \mathbf{J}_{F_2,1}. \tag{109}$$

Thus,

$$\overline{\boldsymbol{W}^{(l)}} = \begin{cases} \boldsymbol{\Omega} \otimes \boldsymbol{W}^{(l)}, & \text{if } l = 1 \\ (\boldsymbol{\Omega} \otimes \boldsymbol{C}^{tr}) \otimes \boldsymbol{W}^{(l)}, & \text{if } l = 2 \end{cases}, \quad \overline{\boldsymbol{B}^{(l)}} = \boldsymbol{\Phi} \otimes \boldsymbol{J}_{F_l,1}. \tag{110}$$

In addition, by setting $L = 2$, we will have $\overline{f_{\overline{\theta}}}(\boldsymbol{r}) = \overline{\boldsymbol{W}^{(3)}} \, \overline{\boldsymbol{z}^{(2)}}$. Note that according to the assumptions of the theorem, $\overline{\boldsymbol{W}^{(3)}} = \boldsymbol{C}^{tr} \otimes \boldsymbol{I}_{F_2}$. As a result, $\overline{f_{\overline{\theta}}}(\boldsymbol{r}) = \overline{\boldsymbol{W}^{(3)}} \, \overline{\boldsymbol{z}^{(2)}} = (\boldsymbol{C}^{tr} \otimes \boldsymbol{I}_{F_2}) \, \overline{\boldsymbol{z}^{(2)}}$, which is equal to $\boldsymbol{z}^{(2)} = f_\theta(\boldsymbol{r})$, as derived in equation 98. equation 98. In conclusion, the theorem holds true for $L = 2$.

Now, suppose that Equation equation 12 holds for $L = 2k$. Consequently,

$$\boldsymbol{z}^{(2k)} = \left(\boldsymbol{C}^{tr} \otimes \boldsymbol{I}_{F_{2k}}\right) \overline{\boldsymbol{z}^{(2k)}} \tag{111}$$

Now, we aim to analyze the case for $L = 2(k+1)$. For this network with two additional layers, we first need to adjust the weight matrix for layer $l = 2k+1$. The new weight matrix will be

$$\overline{\boldsymbol{W}^{(2k+1)}} = \left(\boldsymbol{\Omega} \otimes \boldsymbol{W}^{(2k+1)}\right)\left(\boldsymbol{C}^{tr} \otimes \boldsymbol{I}_{F_{2k}}\right), \tag{112}$$

and the weights and the biases of the two new layers will be

$$\overline{\boldsymbol{W}^{(2k+2)}} = \left(\boldsymbol{\Omega} \otimes \boldsymbol{C}^{tr}\right) \otimes \boldsymbol{W}^{(2k+2)}, \quad \overline{\boldsymbol{B}^{(2k+2)}} = \boldsymbol{\Phi} \otimes \boldsymbol{J}_{F_{2k+2},1},$$
$$\overline{\boldsymbol{W}^{(2k+3)}} = \boldsymbol{C}^{tr} \otimes \boldsymbol{I}_{F_{2k+2}}, \quad \overline{\boldsymbol{B}^{(2k+3)}} = \boldsymbol{\Phi} \otimes \boldsymbol{J}_{F_{2k+3},1}. \tag{113}$$

Now, note that

$$\overline{\boldsymbol{W}^{(2k+1)}} \, \overline{\boldsymbol{z}^{(2k)}} = \left(\boldsymbol{\Omega} \otimes \boldsymbol{W}^{(2k+1)}\right)\left(\boldsymbol{C}^{tr} \otimes \boldsymbol{I}_{F_{2k}}\right) \overline{\boldsymbol{z}^{(2k)}}. \tag{114}$$

Therefore, by setting $l = 2k-1$ in Equation equation 98, or using Equation equation 111, we obtain

$$\overline{\boldsymbol{W}^{(2k+1)}} \, \overline{\boldsymbol{z}^{(2k)}} = \left(\boldsymbol{\Omega} \otimes \boldsymbol{W}^{(2k+1)}\right) \boldsymbol{z}^{(2k)} \tag{115}$$

This is analogous to feeding $\boldsymbol{z}^{(2k)}$ into a neural network whose first layer has the weight matrix $\boldsymbol{\Omega} \otimes \boldsymbol{W}^{(2k+1)}$. Since the additional weight matrices and biases are consistent with Parts 1 and 2 of the proof, we can conclude that

$$\overline{f_{\overline{\theta}}}(\boldsymbol{r}) = \boldsymbol{z}^{(2k+2)} = f_\theta(\boldsymbol{r}). \tag{116}$$

$\square$

## C.3 Proof of Lemma equation 1

*Proof.* Let $[a_{r,1}, a_{r,2}, \ldots, a_{r,T}] \in \mathbb{Q}^T$ be the $r$'th row of $\boldsymbol{\Psi}^{tr}$. Now, define a matrix $\hat{\boldsymbol{A}}$ which is identical to $\boldsymbol{A}$ except for its $r$'th row. This modified row is constructed as follows:

$$\hat{a}_{r,i} = \frac{\sqrt{p_i}}{10^{-\eta}\lfloor 10^\eta \sqrt{p_i}\rfloor} \left(\psi_{r,i} + \epsilon[\psi_{r,i} = 0]\right) \tag{117}$$

in which $p_i$ is the $i$'th prime number, $\epsilon$ is the machine precision, $[.]$ is Iverson bracket, and $\eta$ is a large enough natural number such that $\frac{\sqrt{p_i}}{10^{-\eta}\lfloor 10^\eta \sqrt{p_i}\rfloor} \approx 1$ (to avoid significant changes in the matrix). At the same time, we must have $\left|\frac{\sqrt{p_i}}{10^{-\eta}\lfloor 10^\eta \sqrt{p_i}\rfloor} - 1\right| \geq \epsilon$ (to prevent it from becoming a rational number).

Let $\alpha_i := \frac{\hat{a}_{r,i}}{\sqrt{p_i}}$. Then, $\alpha_i \in \mathbb{Q} \setminus \{0\}$. Now assume that there is $S = [s_1, \ldots, s_T]^{tr} \in Ker(\hat{\boldsymbol{A}}) \cap \mathbb{Q}^T$. Consequently,

$$\sum_{i=1}^{T} \hat{a}_{r,i} s_i = 0 \tag{118}$$

As a result,

$$\sum_{i=1}^{T} \alpha_i \sqrt{p_i} s_i = 0 \tag{119}$$

Note that $\alpha_i s_i \in \mathbb{Q}$. Furthermore, The square roots of all prime numbers are linearly independent over $\mathbb{Q}$ (Stewart, 2022). As a result, $\alpha_i s_i = 0$ for all $i$. Since $\alpha_i \neq 0$, we must have $s_i = 0$ for all $i$, that is, $Ker(\hat{\boldsymbol{A}}) \cap \boldsymbol{Q}^T = \boldsymbol{O}$.[4]                    □

---

[4]Note that all algebraic numbers are computable. This analysis was founded on the computability and expressibility of the square roots of prime numbers in a machine. However, most of the computable numbers are rounded or truncated when stored in a machine. Nevertheless, it is possible to demonstrate theoretically or through simulation that increasing precision can make the aforementioned analysis always feasible.

