# OpenReview forum: "STAF: Sinusoidal Trainable Activation Functions for Implicit Neural Representation"
_ICLR.cc/2025/Conference — ICLR 2025 Conference Withdrawn Submission_

### Official Review · Reviewer_F4LF · 2024-10-25

**Soundness:** 3
**Presentation:** 4
**Contribution:** 4
**Rating:** 3
**Confidence:** 5

**Summary:**

A new approach for INR is proposed where instead of using a single sine or wavelet kind of activation, a Fourier series type of activation with learnable parameters is proposed.  It is shown that STAF is more efficient than previous INRs including SIREN and WIRE.  The authors also present very interesting discussion about some assumptions which were made in SIREN that were wrong.  The mathematics presented in this paper is very clear.  Overall this is a very nicely written paper with a few drawbacks.

**Strengths:**

+ Very well-written paper.
+ Presents the math in a very precise and clear way.
+ Found an important issue in the original SIREN paper which made some wrong assumptions.
+ Efficiency of the proposed overall approach for image representation using STAF.

**Weaknesses:**

- The main limitation of this work is that experiments are very weak. One can not draw a clear conclusion based on what is being presented in the paper.  For example, results are presented only on a handful of images (2-3 images).  It is not clear if the results also hold of different types of images.  Most examples presented in the paper are grayscale kind of images.
Detailed evaluation on many variety of images with proper comparison with other methods will make the paper stronger.  Otherwise, one can not see the significance of this work even though the math is very nice.
- In SIREN, WIRE and many previous INR papers many image restoration experiments were also conducted to see the impact of the INR on such applications.  No such experiments are conducted in this paper.
- Also, no experiments beyond 2-3 images are presented.  What about occupancy fields, NERF etc.?  How well does STAF work on those?

**Questions:**

The main issue with this paper is the lack of experimentation.  Please see my detailed comments above.  Without proper evaluation of STAF on different modalities and image restoration, one can not draw any conclusions about the proposed INR.

**Details Of Ethics Concerns:**

None.

---

### Official Review · Reviewer_j2VX · 2024-10-28

**Soundness:** 4
**Presentation:** 4
**Contribution:** 4
**Rating:** 5
**Confidence:** 4

**Summary:**

The authors introduce Sinusoidal Trainable Activation Functions which reduce spectral bias of Implicit Neural Representations. New initialization scheme with a unique PDF for weight init which helps compared to SIREN. Provide theorems as to why they significantly increase the frequency expressive power of their network. Shows how it improves reconstruction quality compared to given baselines. Essentially the main baseline is SIREN which passes the output of each linear layer through a single sinusoidal, and this paper passes it through a sum of ~25 sinusoidials with scale and shift parameters on the input and output

**Strengths:**

- Strong PSNR increase
- Strong method with minimal parameter overhead. Authors seem to have studied the problem well to come up with this method
- Personal suggestion, will *not* affect my end vote, but i would recommend moving the audio representation into the main paper and the proof of initialization scheme to the end, as the experiments speak for themselves. I find the math provides little to no value, but once again personal preference.

**Weaknesses:**

Baselines feel weak:
- For images there’s SPDER (https://arxiv.org/pdf/2306.15242) which claims to be SOTA for image representation and is very similar, should be compared
- 3D shape representation should have much better baselines than SIREN. Quick online search: DeepSDF, Occupancy Networks, IM-NET. I suggest the authors try to compare to at least 1 of these published in the last 2 years.
- It is not meaningful to keep mentioning that you improve spectral bias over ReLU Networks–I think those in this subfield know that models like SIREN or NeRF are the bare minimum baseline (and there are much more advanced methods actually). I suggest the authors continue focusing on more recent methods.

Tasks:
- Novel view synthesis would be a more meaningful task as it’s more challenging, not sure why it’s not included (i.e. NeRF).
- No video representation? Why were these 2 tasks not included, and do the authors have plans to include them in the future?

Generalizability
- Possibly requires lots of hyperparam tuning? See question 1, where I am curious about how sensitive the model is to hyperparameters such that other researchers can really benefit from this without worrying about grid searching.

**Questions:**

- “though hyperparameter tuning is still required for different tasks.” Can the authors clarify this?
- Why can this not be tested against NeRF/novel view synthesis?
- Why are you training for multiple “epochs”? I thought its full batch gradient descent so it should simply be the number of steps where each gradient step has the entire dataset in it?
- How is Tau determined? Does it vary depending on the task?
- Can you try on other tasks? Video, NeRF, have extremely high information content and could benefit from this research? (see Weaknesses for other baselines/tasks).

---

### Official Review · Reviewer_aJdt · 2024-10-31

**Soundness:** 3
**Presentation:** 3
**Contribution:** 3
**Rating:** 8
**Confidence:** 3

**Summary:**

Implicit Neural Representation (INR) enables representing discrete methods in a continuous manner by utilizing Multi-Layer Perceptrons. For each layer's neurons, the choice of activation function significantly impacts INR's performance. Traditional activation functions (e.g., ReLU) encounter limitations due to spectral bias, which affects the reconstruction of fine details. To address these challenges, the Sinusoidal Trainable Activation Function (STAF) aims to use trainable sinusoidal activation functions. This approach involves not only adjusting the network's weights and biases but also learning the parameters of the activation functions, thereby capturing more accurate signals. STAF employs activation functions that are shared across each layer and combines multiple frequency components into a single parameter set. Additionally, it proposes a stable learning method through an initialization scheme designed to ensure that values within each period follow a standard normal distribution. Through these methods, performance improvements were achieved compared to previous state-of-the-art methods such as WIRE, SIREN, and KAN.

**Strengths:**

1. The choice of activation functions for each layer's neurons significantly impacts the performance of INRs. SIREN demonstrated that periodic activation functions are effective in INRs by utilizing fixed sine functions. In contrast, STAF employs periodic activation functions that are dynamically trained with the MLP's weights and biases during the training process, enhancing the network's ability to better capture and reconstruct high-frequency signals. This design is similar to Fourier series, aiming to achieve maximum efficiency with minimal coefficients. As STAF uses layer-wise sharing, each layer shares a unique activation function, allowing the activation behavior to adjust according to the features processed by each layer. This approach aligns well with the hierarchical structure of MLPs. As a result, STAF has shown superior performance across various representations, including images, audio, and shapes.


2. SIREN's initialization simply uses sine functions to ensure that the output distribution follows a consistent pattern. STAF goes a step further by adjusting the post-activation values to follow a standard normal distribution, thereby enhancing the stability of the training process. This improvement is well-explained with equations and proofs.

**Weaknesses:**

Models utilizing STAF are more efficient than SIREN, as they can achieve higher performance with fewer parameters. This is because the activation function leverages frequency-related parameters to combine multiple sine functions, resulting in greater expressive power. However, as the frequency-related parameter (nf in the code) increases, the computational cost rises rapidly. This can slow down training and require more computational resources and memory. Therefore, when using STAF, it is important to appropriately adjust the nf value to manage the computational burden effectively.

**Questions:**

Minor revision comments:

1. Some commas or periods are missing after equations.
2. On page 7, in the Results and Analysis section, "Figure 8b" is incorrectly referenced and should be "Figure 2."
3. On page 7, in the Results and Analysis section, the description for "Figure 4" is entirely incorrect. The numerical details related to the methods are misplaced.
4. I'm not sure if including NTK (Neural Tangent Kernel) in the key contributions is appropriate, given that it is only discussed in the appendix and not in the main text.

**Details Of Ethics Concerns:**

.

---

### Official Review · Reviewer_DwoS · 2024-11-02

**Soundness:** 3
**Presentation:** 3
**Contribution:** 3
**Rating:** 5
**Confidence:** 4

**Summary:**

The paper (STAF) presents a valuable contribution in designing activation functions for Implicit Neural Representations. STAF is specifically design to handle and reconstruct diverse, complex signals with high accuracy, as clearly reflected in the obtained PSNRs and error plots. STAF utilizes a Fourier series-like approach with trainable amplitude, frequency, and phase, enhancing representation performance compared to static periodic functions like SIREN. Furthermore, the theoretical insights and ideas regarding spectral bias are noteworthy.

**Strengths:**

1). The paper is easy to understand and well written.

2) The proposed approach goes beyond the conventional SIREN architecture through making the activation parameters learnable such that those adapt to the given image, audio or shape.

3). Instead of using one sinusoid at a specific frequency, STAF proposes to use a linear combination of sinusoids (Fourier approach)

4). The mathematical rigorousness of the paper is notable, and STAF aims to prevent issues related of vanishing or exploding gradients during early training phases.

5). The proposed approach has gone in depth to understand the theoretical analysis of INRs

**Weaknesses:**

I believe the core of the paper is good, but I the current version of the paper lacks extensive experimental results. Please see the following parts.

1). The experimental results are only shown for representation tasks (image, occupancy, and audio). If the proposed approach can be extended to inverse tasks that will demonstrate your method's generalization abilities. For instance how STAF works for inverse vision tasks like inpainting or NeRFs.

2). The image representation results have been obtained through down-sampling the cameraman image to 128*128. As down-sampling leads to loss of important features, it might be easy for STAF to approximate the image. Further, it would be great if the paper has results without down-sampling images at least for a couple of Kodak images as in WIRE?

3).I would like to know how the authors initialized the activation function parameters. Each part in the sinusoidal function has three trainable parameters, and the authors used 25 of these for each layer, resulting in 75 trainable parameters per layer. Were these parameters initialized randomly, or were they manually fine-tuned for each image? If they were initialized randomly, do all of them adjusted over epochs, or do only a few converge to a dominant frequency?

4). As this has many trainable parameters for the activation function compared to a single sinusoidal, I would like to know the computational requirement for STAF.

Please address these comments.

**Questions:**

Please see the weaknesses section.

---

### Note · Authors · 2025-01-22

I have read and agree with the venue's withdrawal policy on behalf of myself and my co-authors.